# Learning to Contextualize Web Pages for Enhanced Decision Making by LLM Agents

**Dongjun Lee**[*1]**, Juyong Lee**[*1]**, Kyuyoung Kim**[1]**, Jihoon Tack**[1]
**Jinwoo Shin**[1]**, Yee Whye Teh**[2]**, Kimin Lee**[1]
`{dgjun32, agi.is}@kaist.ac.kr`

KAIST AI[1], University of Oxford[2]

## Abstract

Recent advances in large language models (LLMs) have led to a growing interest in developing LLM-based agents for automating web tasks. However, these agents often struggle with even simple tasks on real-world websites due to their limited capability to understand and process complex web page structures. In this work, we introduce LCoW, a framework for **L**earning language models to **Co**ntextualize complex **W**eb pages into a more comprehensible form, thereby enhancing decision making by LLM agents. LCoW decouples web page understanding from decision making by training a separate contextualization module to transform complex web pages into comprehensible format, which are then utilized by the decision-making agent. We demonstrate that our contextualization module effectively integrates with LLM agents of various scales to significantly enhance their decision-making capabilities in web automation tasks. Notably, LCoW improves the success rates of closed-source LLMs (e.g., Gemini-1.5-flash, GPT-4o, Claude-3.5-Sonnet) by an average of 15.6%, and demonstrates a 23.7% average improvement in success rates for open-source LMs (e.g., Llama-3.1-8B, Llama-3.1-70B) on the WorkArena benchmark. Moreover, the Gemini-1.5-flash agent with LCoW achieves state-of-the-art results on the WebShop benchmark, outperforming human experts. The relevant code materials are available at our project page: https://lcowiclr2025.github.io.

## 1 Introduction

Large language models (LLMs) have demonstrated strong potential in automating web tasks by treating web browsing as a sequential decision-making process, where web pages serve as observations and user interactions, such as clicking and typing, function as actions (Yao et al., 2022a;b). Various approaches have been developed to enhance the performance of LLM agents in these tasks. One such method involves fine-tuning open-source LLMs using demonstration data from web browsing tasks (Furuta et al., 2023; Lai et al., 2024). While promising, this approach requires extensive data collection and significant computational resources for effective fine-tuning. Alternatively, several studies have utilized advanced closed-source LLMs, such as GPT-4o (OpenAI, 2024), with carefully designed prompting techniques (Drouin et al., 2024; Zhou et al., 2023a; Sodhi et al., 2024; Pan

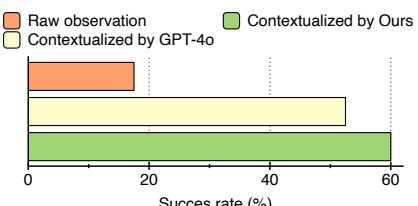

Figure 1: Success rate of the Gemini-1.5-flash agent on 40 WorkArena tasks. We selected a subset of 40 tasks by choosing the first 40 tasks based on the task indices. When the agent leverages observations contextualized by GPT-4o (yellow), its success rate improves by 31%, with further improvements achieved with our method (green).

et al., 2024). By leveraging the general world knowledge and reasoning capabilities of these models, the methods enhance web automation but at the cost of reduced controllability.

However, despite the advancements, state-of-the-art LLM agents often struggle to process complex raw web content, such as HTML and accessibility trees, posing significant challenges for their effec-

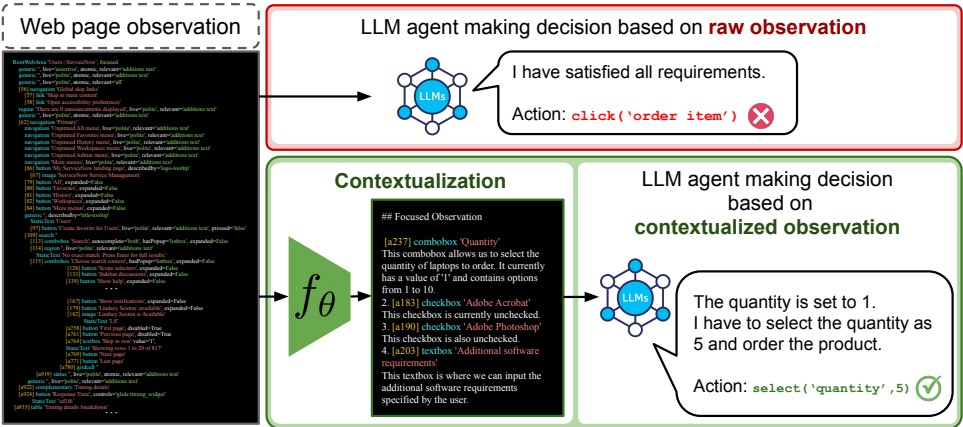

Figure 2: **(Top)** In the conventional pipeline, LLM agents decide on the next action based on raw, complex web page observations (e.g., HTML, accessibility trees), which often hinder accurate decision making. **(Bottom)** In our proposed pipeline, a contextualization module transforms these complex web page observations into a more comprehensible format, thereby enabling LLM agents to make more accurate decisions by enhancing their understanding of the web page.

tive use in web task automation. While LLMs excel in tasks that require detailed reasoning, such as solving mathematical problems or coding, we hypothesize that their underperformance in seemingly simple decision-making tasks like web browsing is not due to a lack of decision-making capabilities but rather to difficulties in understanding and processing complex web page observations.

To validate our hypothesis, we conducted an initial experiment that demonstrated an LLM agent based on Gemini-1.5-flash can achieve substantial improvements in web browsing tasks when equipped with a module designed to contextualize complex web page observations (i.e., contextualization module). This module enhances task performance by removing irrelevant UI elements and highlighting key components with explanations, thereby simplifying the decision-making process. We evaluated the performance of this agent on 40 tasks from WorkArena (Drouin et al., 2024), a benchmark designed to assess web agents on real-world, enterprise-related websites. As shown in Figure 1, utilizing GPT-4o as the contextualization module (yellow) resulted in a 31% absolute improvement in the agent's success rate compared to direct processing of raw observations (red). These results support our hypothesis that the difficulty in understanding web pages is a major bottleneck for LLM-based web agents.

In this work, we propose LCoW, a framework that includes a contextualization module and a training algorithm to fine-tune this module to enhance the decision-making capabilities of LLM agents in web automation. As illustrated in Figure 2, the contextualization module transforms complex web page observations into a comprehensible format, enabling LLM agents to make more accurate decisions. Furthermore, to enable the contextualization module to provide context more grounded in real websites, we propose an iterative algorithm designed to train the contextualization module. Our training algorithm consists of three phases: (i) trajectory collection, (ii) sampling contextualized observations, and (iii) updating the contextualization module. For each observation from the collected trajectories, we generate multiple contextualized observations using the current contextualization module. Each observation is then assigned a reward based on whether a set of LLM agents can accurately predict the correct action given the contextualized observation. Finally, we select the one with the highest reward as the target and train the contextualization module to maximize the likelihood of the target given the original raw observation.

As demonstrated in our initial experiment using the Gemini-1.5-flash (see Figure 1), LCoW significantly enhances the decision-making capabilities of LLM agents, even beyond the improvements seen with state-of-the-art LLMs like GPT-4o used as a contextualization module. In our experiments, we conduct comprehensive evaluations of our proposed approach on three well-established benchmarks for evaluating agent performance in web environments: WebShop (Yao et al., 2022a), WorkArena (Drouin et al., 2024), and WebArena (Zhou et al., 2023a). In detail, we first demonstrate that LCoW significantly enhances the overall performance of LLM agents with varying scales in the main experiment. Furthermore, we analyze the behavior of the contextualization module in

detail, including qualitative examination of the observation refined by the contextualization module, comparison with training LLM agents using behavior cloning, and generalization ability.

## 2 BACKGROUND

In this section, we describe the formulation of web browsing as a sequential decision-making problem and the use of LLMs as decision-making agents.

Web browsing can be formulated as a Partially Observable Markov Decision Process (POMDP), defined by $\langle \mathcal{S}, \mathcal{O}, \mathcal{A}, T, \mathcal{R} \rangle$. The state $s_t \in \mathcal{S}$ represents the internal configuration of the web browser at time step $t$, which is only partially observable. The observation $o_t \in \mathcal{O}$ corresponds to the web page rendered by the web browser given $s_t$, which can take various forms (e.g., screenshot, HTML, accessibility trees). The action space $\mathcal{A}$ is the set of all possible interactions with the UI elements (e.g., clicking, typing). The state transition function $T$ defines the probability of transitioning from state $s$ to state $s'$ after performing an action $a_t \in \mathcal{A}$, such as clicking a link or scrolling. While transitions are typically deterministic, occasional stochastic events (e.g., pop-ups, network errors) can occur. The reward function $\mathcal{R}$ assesses the functional correctness, evaluating whether the resulting state $s_t$ satisfies pre-defined criteria for successful task completion.

Leveraging their ability to interpret web pages and generate actions in text form, LLMs are increasingly employed as agents for automating web-based tasks. In this framework, an LLM agent $\pi$ generates an action $a_t$ to interact with UI elements at each time step $t$, based on the user instruction [TASK], the current web page observation $o_t$, and the history of previous actions $a_{<t}$. The objective of the agent is to complete the given task in order to maximize the reward.

## 3 METHOD

In this section, we present LCoW, a framework for enhancing the capability of LLM agents by contextualizing complex web pages. Section 3.1 outlines the concept of the contextualization module and its integration with LLM agents for web task automation. Section 3.2 introduces an iterative algorithm for training the contextualization modules to improve decision making of LLM agents.

### 3.1 CONTEXTUALIZATION MODULE

In this work, we decouple web page understanding from decision making of LLM agents. Our hypothesis is that while LLM agents possess strong decision-making capabilities, their performance can significantly degrade when they rely on lengthy, non-contextualized observations, such as HTML and accessibility trees. To address this limitation, we introduce a *contextualization module*, a separate language model designed to enhance LLM agents by contextualizing complex web page observations into a form that is easier to process and comprehensible. Intuitively, a proper contextualization of observations can enhance the agent's understanding of web content and its decision making based on the improved understanding (see Figure 3 for an example of the input and output of the module). Formally, given a web page observation $o_t$ at time step $t$, the objective of our contextualization module $f_\theta$ is to generate a contextualized observation $o_t^{\text{co}}$ that serves as input to the LLM agent $\pi$ to enhance its decision making. Specifically, $f_\theta$ uses the task instruction [TASK], the previous actions of the agent $a_{<t}$, and the current web page observation $o_t$ to generate a contextualized observation:

$$o_t^{\text{co}} = f_\theta(\texttt{[TASK]}, a_{<t}, o_t).$$

The LLM agent $\pi$ then predicts the next action based on the contextualized observation:

$$a_t = \pi(\texttt{[TASK]}, a_{<t}, o_t^{\text{co}}).$$

While an arbitrary language model can serve as a contextualization module $f_\theta$, it is important for the module to learn from experience in the web environment to provide more grounded context for decision making, such as role of a particular button or interaction with specific UI elements.

### 3.2 ALGORITHM FOR TRAINING THE CONTEXTUALIZATION MODULE

We now describe an iterative algorithm for training the contextualization module $f_\theta$ to enhance decision making of LLM agents. In a nutshell, the algorithm involves an iterative process of col-

```
Role: Your task is to generate a "Reasoning" and a "Refined      Reasoning: We have navigated to the Hardware store and
observation" based on the provided information.                  clicked on the iPad pro link. The actions needed to accomplish
                                                                 the user instruction are to choose color option and disk option.
Task: "Purchase me an iPad pro with silver color and ..."
Action history:                                                  Refined observation: I can focus on following elements:
 1. click [a32] ''Hardware"
 2. click [a45] ''iPad pro"                                      [a192] combobox 'Quantity' value='1', checked=true,
Web page observation:                                            It confirms that 1 is already selected as a quantity.

RootWebArea 'Catalog | ServiceNow'                               [a195] radiogroup 'Mandatory'
    generic '', live='assertive', ...                                [a196] radio '\\uf Silver', checked=True
    generic '', live='polite', ...                                   [a197] radio '\\uf Space Gray'
    [55] navigation 'Global skip links'                          This group contains the color options.
    ...
    [a157] rowgroup ''                                           [a198] radiogroup 'Mandatory'
        [a158] row ''                                                [a200] radio '\\uf137 128 GB', checked=True
            [a159] gridcell ''                                       [a203] radio '\\uf137 256 GB [add $100.00]'
            ...                                                  This group contains the storage options.
    [a401] LayoutTableRow ''
        [a402] LayoutTableCell 'Shopping Cart'                   [a206] button 'Order Now'
            [a403] LayoutTableCell 'Empty'                       This button allows to finalize the process.
```

Figure 3: An example of a input of contextualization module including lengthy web page observation **(left)** and an observation contextualized by the contextualization module trained using LCoW **(right)**. The module converts raw observations into a more concise form to enhance decision making in agents. The prompt used is provided in Appendix C.

lecting paired input-output data for training the contextualization module and subsequently updating the module based on the collected data. For data collection, we begin by gathering trajectories of successfully completed tasks from the web browsing environment. For each observation $o_t$ in the collected trajectories, we sample multiple candidate contextualized observations, and select the one that best provides the relevant context for multiple LLM agents to accurately predict the next action $a_t$. Based on the chosen target observations, we update $f_\theta$ via supervised fine-tuning. We now outline a single iteration of LCoW, followed by a detailed explanation of the design of the reward used for evaluating the candidate contextualized observations.

**Single iteration** A single iteration of LCoW starts with the contextualization module $f_{\theta(i)}$ and aims to update this module to $f_{\theta(i+1)}$. This process consists of three steps:

**Step 1 (Trajectory collection).** Given a set of training tasks, we roll out the LLM agent $\pi$ in the web environment to collect trajectory data. Specifically, the agent determines the next action based on the contextualized observation produced by $f_{\theta(i)}$ until the episode terminates. We collect only those trajectories that end in the successful completion of the tasks.

**Step 2 (Sampling optimal contextualization).** As illustrated in Figure 4, we start by sampling multiple candidates $o_t^{co}$ from the current contextualization module $f_{\theta(i)}$ (i.e., $o_t^{co} \sim f_{\theta(i)}(\cdot \mid [\text{TASK}], a_{<t}, o_t)$) for each web page observation $o_t$ in the collected trajectories. Each candidate is then assigned a reward based on whether a set of LLM agents can accurately predict the ground-truth action $a_t$ given $o_t$, with the candidate receiving the maximum reward selected as the optimal contextualized observation. If all candidates receive a zero reward, we retry the sampling process with the ground-truth action $a_t$ provided as additional context to $f_{\theta(i)}$ (i.e., $o_t^{co} \sim f_{\theta(i)}(\cdot \mid [\text{TASK}], a_{<t}, o_t, a_t)$) to guide the generation of valid contextualized observations.

**Step 3 (Model update).** We update the current module $f_{\theta(i)}$ by fine-tuning it with the optimal contextualized observations collected in Step 2. With the updated module $f_{\theta(i+1)}$, we return to Step 1 and repeat the process.

Algorithm 1 outlines the steps for a single iteration of the training algorithm. Starting with an initial contextualization module $f_{\theta(0)}$, we iteratively train the module through $M$ iterations. After each iteration, the module updates from $f_{\theta(i)}$ to $f_{\theta(i+1)}$ until reaching the final $f_{\theta(M)}$.[1] Additionally, $\mathcal{T}$ can be initialized as a set of human demonstrations in order to facilitate the training process.

**Reward for contextualized observations** The reward for the contextualized observation $o_t^{co}$ is defined as the sum of the action-matching scores computed using multiple LLM agents. Each action matching score evaluates whether an LLM agent $\pi$ correctly predicts the ground-truth action $a_t$ given the contextualized observation $o_t^{co}$. By leveraging multiple LLM agents to compute the reward, we

---

[1]In the initial iteration, we assume a limited set of seed demonstrations and use relatively strong LLMs to sample candidate contextualized observations to accelerate training.

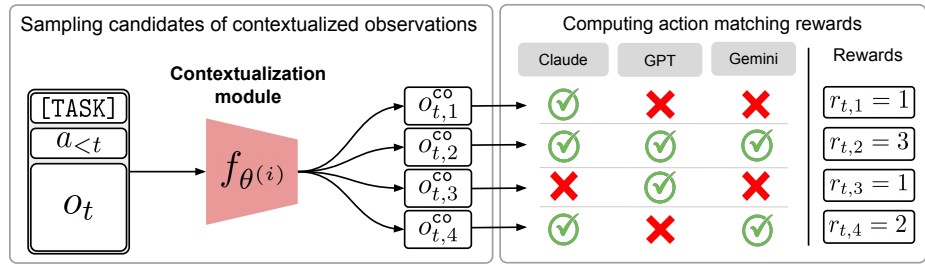

Figure 4: Illustration of sampling optimal contextualization. First, we sample multiple candidates of contextualized observations, given user instruction $[\text{TASK}]$, previous actions $a_{<t}$, and observation $o_t$. Subsequently, multiple LLM agents predict the next action based on each candidate, and the reward for each candidate is computed according to how many LLM agents correctly predict the ground-truth action $a_t$.

---

**Algorithm 1** One iteration of LCoW

---

**Require:** a contextualization module $f_{\theta^{(i)}}$, an LLM agent $\pi$, a set of LLM agents for computing the reward $\Pi = \{\pi_i\}_{i=1}^{K}$, a trajectory buffer $\mathcal{T}$, an empty data buffer $\mathcal{D}$, and a set of training tasks $\mathcal{G}_{tr}$

1: // Trajectory collection
2: **for** $[\text{TASK}] \in \mathcal{G}_{tr}$ **do**          ▷ Collect successful trajectories from training environment
3:      $\tau, R \sim \pi(\cdot \mid [\text{TASK}], f_{\theta^{(i)}})$
4:      **if** $R = 1.0$ **then**
5:          $\mathcal{T}$.append($\tau$)
6: // Sampling optimal contextualizations
7: **for** $(o_t, a_t)$ in $\mathcal{T}$ **do**
8:      **for** $n \leftarrow 1$ to $N$ **do**    ▷ Sample $N$ candidate contextualized observations and assign rewards
9:          $o_{t,n}^{\text{co}} \sim f_{\theta^{(i)}}(\cdot \mid [\text{TASK}], a_{<t}, o_t)$
10:          $r_{t,n} = \sum_{\pi \in \Pi} \text{ActionMatchingScore}(\pi([\text{TASK}], a_{<t}, o_{t,n}^{\text{co}}), a_t)$
11:      **if** $\max_n(r_{t,n}) = 0$ **then**
12:          **for** $n \leftarrow 1$ to $N$ **do**          ▷ Retry the sampling if rewards are zero for all candidates
13:             $o_{t,n}^{\text{co}} \sim f_{\theta^{(i)}}(\cdot \mid [\text{TASK}], a_{<t}, o_t, a_t)$
14:             $r_{t,n} = \sum_{\pi \in \Pi} \text{ActionMatchingScore}(\pi([\text{TASK}], a_{<t}, o_{t,n}^{\text{co}}), a_t)$
15:      $o_{t,*}^{\text{co}} = \arg\max_n(r_{t,n})$
16:      $\mathcal{D}$.append($[([\text{TASK}], a_{<t}, o_t), o_{t,*}^{\text{co}}]$)
17: // Parameter update
18: $\theta^{(i+1)} := \arg\max_{\theta^{(i)}} \mathbb{E}_{([\text{TASK}], o_{t,*}^{\text{co}}, a_{<t}, o_t) \sim \mathcal{D}}[f_{\theta^{(i)}}(o_{t,*}^{\text{co}} \mid [\text{TASK}], a_{<t}, o_t)]$
19: **return** $f_{\theta^{(i+1)}}$

---

ensure that the module produces contextualized observations that generalize across a diverse set of agents, preventing overfitting to the behavior of any single agent and enhancing the module's adaptability to arbitrary LLM agents.

## 4 EXPERIMENTS

We design our experiments to investigate the following questions:

- How effective is LCoW in training a contextualization module for improving decision making of LLM agents? (Section 4.2)
- Can the contextualization module trained with LCoW generalize to arbitrary LLMs with varying scales? (Section 4.2)
- What form do the web page observations take after contextualization and how the contextualized observation aid the decision making of LLM agents? (Section 4.3)
- Can the contextualization module trained with LCoW generalize to unseen tasks? (Section 4.3)

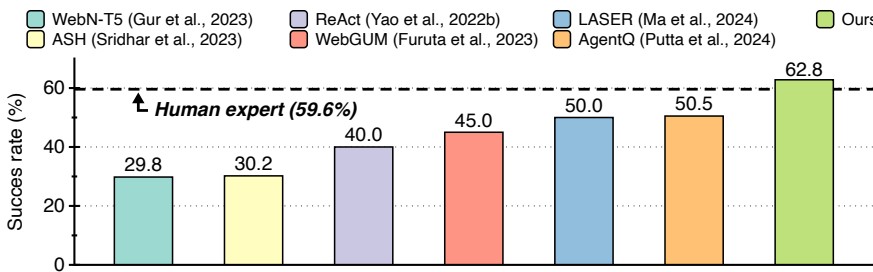

Figure 5: Success rate on 500 evaluation tasks from WebShop. Average human performance and expert human performance are 50% and 59.6%, respectively (Yao et al., 2022a). The Gemini-1.5-flash agent with the contextualization module trained for three iterations achieves a state-of-the-art success rate of 62.8%, outperforming the human expert performance, as well as previous baselines (Yao et al., 2022b; Furuta et al., 2023; Putta et al., 2024; Sridhar et al., 2023; Ma et al., 2024; Gur et al., 2023).

## 4.1 BENCHMARKS & EVALUATION SETUP

In this section, we desribe three benchmarks used for our experiments. We consider the three benchmarks, WebShop (Yao et al., 2022a), WorkArena (Drouin et al., 2024), and WebArena (Zhou et al., 2023a), which are well-established benchmarks designed to evaluate the capabilities of web agents in completing various web tasks. For the main experiment, we exploit Webshop and WorkArena. For the additional analysis, we also conduct experiments in WebArena. Additional details about hyperparameters and prompts are described in Appendix B and Appendix C, respectively.

**WebShop**  WebShop provides a simulated online shopping environment with real-world product data, consisting of 500 evaluation tasks and 5,500 training tasks, each defined by natural language instructions to purchase products that meet specific criteria. During each decision-making step, the agent receives accessibility tree formatted web page observation to predict the next action, and at the end of an episode, it receives a reward ranging from 0 to 1 based on how well the attributes of the purchased item align with the intended product criteria. When the reward is equal to 1, the episode is considered to be a success. In our experiment, we train the contextualization module using LCoW on environments associated with 500 of the 5,500 training tasks in WebShop. After training, we evaluate the module on 500 evaluation tasks, measuring both the success rate and average reward.

**WorkArena**  WorkArena focuses on assessing web agents' ability to complete enterprise-related tasks, such as creating user accounts and ordering products from a service catalog, on realistic websites. This benchmark comprises 33 task types, with each task type containing up to 1,000 individual task instances, where agents have to navigate over web pages by following natural language instructions and receive rewards based on successful task completion. The agent receives binary rewards (i.e., 0 or 1) based on whether the instruction is achieved via web navigation. In our experiment, we train the contextualization module on 15 task instances for each of 33 task types (i.e., 495 tasks as a total) and evaluate the success rate on 5 task instances for each of 33 task types (i.e., 165 evaluation tasks).

**WebArena**  WebArena consists of 812 tasks spanning over 6 websites (Shopping, Gitlab, Reddit, Map, Wikipedia, and content management systems), where 812 individual tasks are created based on 190 task templates. Similar to WorkArena, WebArena is also based on a realistic web environment, which incorporates extremely long and complex web page observation. For each task, defined as a natural language instruction, the agent receives binary rewards based on whether the instruction is achieved by web navigation. Additionally, Liu et al. (2024) proposed 165 evaluation tasks filtered from the original 812 tasks in WebArena, for efficient evaluation. For the experiment, we evaluate the success rate on these 165 tasks while utilizing the remaining 647 tasks in the original WebArena benchmark as training tasks.

| | GPT-4o | | Gemini-1.5-flash | | Claude-3.5-Sonnet | | Llama-3.1-70B (Unseen) | |
|---|---|---|---|---|---|---|---|---|
| | Success | Reward | Success | Reward | Success | Reward | Success | Reward |
| Raw observation | 34.8% | 0.496 | 43.6% | 0.693 | 26.6% | 0.336 | 34.2% | 0.590 |
| Self-ctx | 26.2% | 0.459 | 46.4% | 0.608 | 12.4% | 0.146 | 40.2% | 0.547 |
| LCoW (iter 1) | 27.8% | 0.545 | 46.4% | 0.705 | 39.4% | 0.600 | 39.2% | 0.666 |
| LCoW (iter 2) | 46.0% | 0.647 | 58.2% | 0.796 | 58.8% | **0.780** | 55.0% | 0.781 |
| LCoW (iter 3) | **50.6%** | **0.666** | **62.8%** | **0.803** | 59.8% | 0.771 | **59.6%** | **0.803** |

Table 1: We investigate the efficacy of LCoW across multiple LLM agents in WebShop. For all LLM agents, LCoW consistently improves both success rate and reward over iterations, surpassing self-contextualization (self-ctx) and even human expert-level success rate by the third iteration. Additionally, LCoW is also effective when combined with Llama-3.1-70B, which was not used for computing the action-matching reward (i.e., unseen) during training the contextualization module.

| | GPT-4o | Gemini-1.5-flash | Claude-3.5-Sonnet | Llama-3.1-70B (Unseen) | Llama-3.1-8B (Unseen) |
|---|---|---|---|---|---|
| Raw observation | 38.2% | 11.5% | 44.8% | 26.1% | 1.2% |
| Self-ctx | 43.0% | 12.7% | 50.3% | 29.1% | 7.3% |
| LCoW (iter 1) | **44.2%** | **41.2%** | **55.8%** | **40.0%** | **37.0%** |

Table 2: We evaluate the success rate of five LLM agents with varying scales on 165 tasks in the WorkArena benchmark. Single iteration of LCoW shows improvement of success rate over all LLMs, even generalized to Llama-3.1-70B and Llama-3.1-8B, which were not used for computing the action matching reward.

## 4.2 MAIN RESULTS

We first demonstrate the effectiveness of LCoW on WebShop and WorkArena benchmarks and show that the contextualization module trained via LCoW enhances performance even for an LLM agent that was not involved in the LCoW training process, such as Llama-3.1-70B. We evaluate against two baselines: (i) decision making based solely on raw observations without the contextualization module (i.e., Raw observation), and (ii) decision making based on the observation contextualized by the LLM agent itself (i.e., self-contextualization).

**Effectiveness of LCoW** As shown in Table 1, in WehShop benchmark, both the success rates and the average rewards achieved by all three LLM agents improve substantially when integrated with the contextualization module trained with LCoW. Particularly, both the Gemini-1.5-flash and Claude-3.5-Sonnet agents surpass the average human performance of 50.5% when combined with the contextualization module trained for 2 iterations. When the contextualization module is trained for 3 iterations, the agents exceed the expert human-level performance of 59.6%. As illustrated in Figure 5, the Gemini-1.5-flash agent combined with LCoW achieves state-of-art performance on the WebShop benchmark, outperforming prior methods in success rate by more than 12%. Notably, the Claude-3.5-Sonnet agent, lower performing than the other agents on WebShop, achieves 59.8% of success rate when integrated with our contextualization module, demonstrating that LCoW can effectively enhance the decision-making capabilities of LLM agents.

As shown in Table 2, on WorkArena, the Claude-3.5-Sonnet agent outperforms the GPT-4o agent, achieving a success rate of 44.8% in our evaluation setup. When integrated with LCoW, Claude-3.5-Sonnet agent achieves an even higher success rate of 55.8%, which is higher than all baselines evaluated. GPT-4o and Gemini-1.5-flash also improve 5% and 30%, respectively, when combined with LCoW.

**Generalization to arbitrary LLM agents** We evaluate the Llama-3.1-8B agent and Llama-3.1-70B agent integrated with LCoW to assess its effectiveness with LLM agents not involved in the training process (i.e., those not used for computing action-matching rewards). As shown in Table 1, the Llama-3.1-70B agent combined with the contextualization module trained for two iterations outperforms average human performance (50.0%) on WebShop and achieves the success rate of human experts (59.6%) when trained for three iterations. Furthermore, Table 2 shows that the contextualization module also enhances Llama-3.1-8B and Llama-3.1-70B agent on WorkArena, improving

```
Contextualized observation:                    Contextualized observation:
The AXTree observation shows we are on the     The AXTree observation shows a list of forums on a
Shipping Report page. The page contains a      website called Postmill. Each forum is presented as an
filter section with date range inputs,         article with its name, number of subscribers, and number
order status options, and other report         of submissions. The forums are sorted by the number of
parameters. There's also an export function    submissions in descending order.
and a table to display the report results.                        ...
                  ...                           Additionally, for context, we can include the first forum
Following elements are crucial for completing  in the list to show where we are currently:
the user instruction:                          [141] article ''
        [753] LabelText ''                          [143] heading 'AskReddit — AskReddit'
            StaticText 'From'                            [144] link 'AskReddit — AskReddit'
            StaticText '*'                          [149] paragraph ''
        [756] textbox 'From *' value='08/05/2022'  [152] Section ''
        [757] button 'undefined \ue627'                 [154] button 'Subscribe No subscribers'
        [760] LabelText ''                                   StaticText '0'
            StaticText 'To'                             [160] paragraph ''
            StaticText '*'                                   StaticText '10,041 submissions'
        [763] textbox 'To *' value='03/01/2023', focused  This extraction shows the first forum in the list, which
        [764] button 'undefined \ue627'            is "AskReddit" with 10,041 submissions. It confirms that
This section contains the filter parameters    we are currently on the first page of forums and need to
for the shipping report. The date range        navigate to the next page to continue our search for the
inputs (From and To) have already been         UpliftingNews subreddit.
filled with the specified dates (08/05/2022
and 03/01/2023, respectively).
```

Figure 6: Examples of how the contextualization module refines complicated web pages into comprehensible format from WebArena benchmark (**Left**) and WorkArena benchmark (**Right**). As indicated by blue color, the contextualization module provides comprehensible context by verbally explaining the web page and UI elements relevant to the given task.

the success rate by approximately 36% and 13%, respectively. It is noteworthy that a relatively small LLM agent (i.e., Llama 3.1-8B) struggles to perform tasks when given raw observations, but when combined with LCoW, its success rate rises on significant margin. It demonstrates that LCoW can elicit a significant level of decision-making capability even from smaller models at the 8B scale.

## 4.3 ANALYSIS

**How web pages are contextualized?** We qualitatively analyze how the contextualization module, trained via LCoW, processes real-world web page observations and how the contextualization enhance decision making of LLM agents on web tasks. As shown in Figure 6, we observe that the contextualization module not only simplifies the web page by extracting UI elements relevant to the given task, but also verbally explains functionalities of the UI elements in web pages. These explanations aid LLM agents significantly in making more accurate decisions, while reducing the selection of inadmissible or redundant actions, which is the main failure mode of the LLM agent for web browsing. This knowledge of the functionality and interaction of UI elements grounded to the real website appears to have been learned during the LCoW training process, where the contextualization module explores various observation contextualizations and learns to generate contextualized observations that lead to more accurate decision making by LLM agents. More detailed examples and analyses are provided in the Appendix A.

**Comparison to directly training LLM agents** One might argue that, given available demonstrations, it would be more straightforward to train the LLM agent directly rather than using the demonstrations to train a contextualization module. To demonstrate the superiority of training the contextualization module with an equivalent amount of demonstration data, we conduct comparative experiments against directly training LLM agent. In this analysis, we fine-tune Llama-3.1-8B with 264 seed demonstrations, the same demonstrations we used for training the contextualization module, using behavior cloning (BC) as a baseline. To ensure a fair comparison in terms of model scale, we define a Llama-3.1-8B agent that performs tasks based on the output of a contextualization module trained using LCoW as the direct comparison group. As shown in Figure 7, both smaller LLM agents, such as Llama-3.1-8B, and larger LLM agents achieve higher success rates when using the LCoW-trained contextualization module compared to the BC baseline.

**Generalization to tasks with unseen type** Furthermore, we evaluate LCoW's generalization capabilities in WebArena benchmark. Specifically, we assess whether the contextualization module trained on specific task types generalizes to tasks belonging to task types unseen during the train-

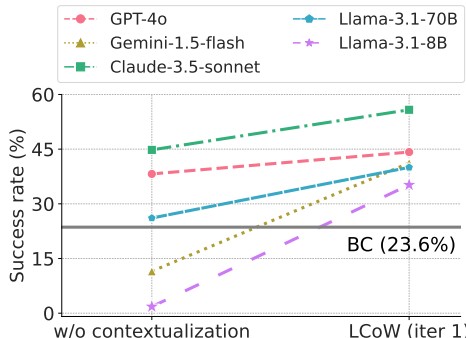

Figure 7: Comparison to training LLM agent via behavior cloning (BC) in WorkArena. We fine-tune Llama-3.1-8B with demonstrations as a BC baseline.

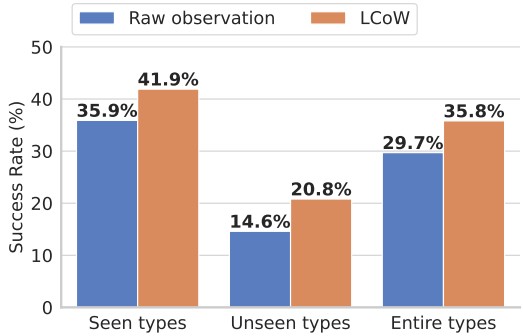

Figure 8: Comparison of GPT-4o agent navigating web pages based on raw observation and observation contextualized by LCoW in WebArena.

ing. Details on the definition of unseen task types are provided in the Appendix B.2. Among 165 evaluation tasks in WebArena, 117 tasks belong to the seen task types, and the remaining 48 tasks belong to the unseen task types. As shown in Figure 8, LCoW consistently shows 6% improvement in the success rate on both seen task types and unseen task types, compared to the GPT-4o agent without LCoW. We expect that generalization to unseen task types occurs due to the fact that the contextualization module learns to explain the UI elements in the web pages (e.g., the role of UI elements in complex web page observation) via LCoW-training, which enhances accurate decision making of LLM agents even in the unseen-type tasks. We also provide additional studies on the generalization ability of LCoW with the WorkArena benchmark in Appendix B.3.

**Can LCoW optimize the contextualization module for specific LLM agents?** We demonstrate that LCoW effectively optimizes the contextualization module to generate context that leads to more accurate decision-making of LLM agents. In the WebShop benchmark, we used the contextualization module checkpoint from each LCoW training iteration to generate contextualized observations for the 1,372 raw observations present in the 397 seed demonstrations. We then calculated the action-matching reward by comparing the actions predicted from these contextualized observations with the ground-truth actions derived from the demonstrations. Figure 9 shows the average action-matching reward across three iterations, demonstrating that with each round of LCoW-training, the contextualization module increasingly produce contextualized observations that enhance decision making of the LLM agents used in data collection (i.e., GPT-4o, Gemini-1.5-flash, Claude-3.5-Sonnet). This shows the effectiveness of LCoW in optimizing the contextualization module for specific LLM agents. Furthermore, it suggests the potential to extend LCoW as a method for indirectly optimizing the behavior of closed-source LLM agents, where direct optimization is not feasible, across a wide range of text-based decision-making tasks.

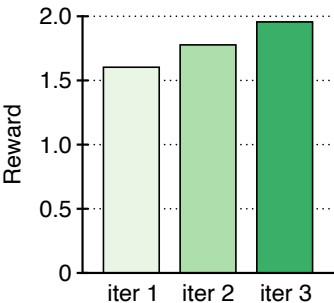

Figure 9: Average action-matching rewards across three iterations on WebShop show a consistent increase, suggesting that the contextualization module is optimized to generate observations that enhance decision-making in LLM agents.

## 5 RELATED WORK

**LLM agents for web automation** As LLMs continue to improve and demonstrate remarkable performance across diverse domains, developing web automation agents based on LLMs is gaining growing interest (Zhou et al., 2023a; Drouin et al., 2024). Recent works have explored various methods to utilize LLMs as agents for automating real-world web tasks. Pan et al. (2024) propose to apply a self-refine mechanism (Madaan et al., 2023) to improve decision making of agents through

self-generated feedback. Sodhi et al. (2024) enhance web automation by using LLMs to manage low-level workflows handcrafted by humans, while Wang et al. (2024) extend this approach by introducing agent workflow memory, which extracts reusable routines from past experiences.

Similar to our work, a promising direction for enhancing web automation involves integrating a summarization module that condenses web page observations, enabling agents to predict actions based on these summarized inputs. For example, Deng et al. (2024) propose MindAct, which consists of an HTML extraction module and an action prediction module. The HTML extraction module ranks individual HTML elements using a ranking language model trained to assess relevance based on user instructions and past actions. Additionally, Gur et al. (2024) introduce HTML-T5, a specialized language model for web page understanding pre-trained on a large corpus of HTML documents and fine-tuned for extractive summarization.

**Automated prompting for closed-source models**  The quality of outputs from closed-source LLMs heavily depends on the prompts used, leading to a substantial body of research dedicated to prompt engineering to elicit more effective responses from these models. (Kojima et al., 2022; Yao et al., 2022b; Lightman et al., 2024).  For example, several recent studies have explored automating the discovery of more effective prompting formats (Shin et al., 2020; Zhou et al., 2023b). Mañas et al. (2024) demonstrate that prompt refinement can yield more consistent responses from closed-source models, especially with text-to-image models. Also, Xu et al. (2024) have suggested compressing the input content for LLMs, focusing on the task of retrieval-augmented generation.

## 6 LIMITATIONS AND FUTURE DIRECTIONS

**Dataset collection procedure**  We leveraged successful trajectories during training when using LCoW in our experiment. We acknowledge that this can be a bottleneck in learning completely new tasks that are not covered by the collected successful trajectories. To overcome this, incorporating an LLM-based search algorithm in the trajectory collection phase to gather successful trajectories for more challenging tasks can be an interesting future direction.

**Contextualization cost**  LCoW incurs latency in the decision-making process due to the computational cost required to generate the contextualized observation. Considering that most tokens of contextualized observation correspond to extraction from the raw observation, we expect LCoW can be integrated with an efficient decoding strategy (e.g., speculative decoding) for resolving the latency problem.

**Generalization ability**  Although we find that LCoW demonstrates generalization in several cases (e.g., unseen task types or unseen website), we acknowledge that we did not observe generalization to a broader range of tasks incorporating various unseen UI elements (see Appendix B.3 for more details), due to limited scale of experiments. We believe that scaling the dataset for the training contextualization module can be a promising direction for better generalization. For example, we expect leveraging large-scale web browsing demonstrations covering a wide range of websites as seed demonstrations would improve generalization to a broader range of tasks and websites.

## 7 CONCLUSION

In this work, we introduce LCoW, a novel approach to enhancing LLM agents in completing web tasks by leveraging language models to contextualize complex web pages into a more comprehensible form. Our approach separates the understanding of web content from the decision-making process by training a specialized module that generates contextualized representations of complex web pages, which are utilized by LLM agents for enhanced decision making. Through extensive experiments, we demonstrate that this contextualization module significantly improves the decision-making capabilities of LLM agents of varying scales.

## ETHICS STATEMENT

We introduce LCoW, a framework for improving decision-making capability of LLM agents for web automation. We caution that LLM agents may cause safety issues such as cybersecurity or risks regarding private information, while we believe that LCoW can be valuable for guiding the agents from potential mistakes by providing more contextualized information on the UI elements. Additionally, we believe the improved efficiency and capability of LLM agents with our methods can provide social opportunities to improve user interactions by using digital devices for those with disabilities.

## REPRODUCIBILITY STATEMENT

For the reproducibility of our results, we have provided a detailed description of our experimental setups and prompts in Section 4.1, Appendix B, and Appendix C. Across the entire experiment, we set the temperature hyperparameter of backbone LLM as 0.0 to enable reproduction. Additionally, to further facilitate reproduction, we open-source our work.

## ACKNOWLEDGMENTS

This research was partly supported by the MSIT(Ministry of Science, ICT), Korea, under the Global Research Support Program in the Digital Field program)(RS-2024-00436680, 70%) supervised by the IITP(Institute for Information & Communications Technology Planning & Evaluation). This work was partly supported by Institute for Information & communications Technology Planning & Evaluation(IITP) grant funded by the Korea government(MSIT) (RS-2019-II190075, Artificial Intelligence Graduate School Program(KAIST), 10%). This work was partly supported by the National Research Foundation of Korea(NRF) grant funded by the Korea government(MSIT) (No. RS-2024-00414822, 10%). This work was partly supported by the Institute of Information & Communications Technology Planning & Evaluation(IITP)-ITRC(Information Technology Research Center) grant funded by the Korea government(MSIT)(IITP-2025-RS-2024-00436857, 10%)

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

# A QUALITATIVE ANALYSIS

In this section, we first analyze (1) examples of contextualized real-world web pages and (2) the decision-making process of LLM agents with and without LCoW.

## A.1 EXAMPLES OF CONTEXTUALIZED WEB PAGES

We provide 5 examples of contextualized web page observations, along with task instructions. As demonstrated in the following examples, the contextualization model trained via LCoW not only summarizes lengthy web pages but also provides contexts (e.g., what happens if a certain UI element is clicked) that enable more accurate decision-making of LLM-based web agents.

- **Example 1**. The contextualized observation provides the necessary information required for sorting the list, including several column headers of the spreadsheet. Especially, it explains the "Personalize List" UI element, which initiates sorting. In many cases, LLM agents without contextualization do not click the "Personalize List" button due to a lack of knowledge about this UI element.

- **Example 2**. This example is drawn from the unseen-type tasks in WebArena, where the GPT-4o agent with LCoW succeeded, while the GPT-4o agent with raw observation failed. In this example, the contextualization module correctly extracts the rows corresponding to the latest order from the raw observation, thereby informing the GPT-4o agent of the status of the latest order. However, without LCoW, due to complex raw observation corresponding to this contextualized version, GPT-4o agent fails to identify the latest order. Although this task belongs to an unseen-type task, as table format frequently appears in web page observations in the training tasks, the contextualization module correctly extracts content from complex web pages, thereby improving the decision-making of LLM agents.

- **Example 3**. This example is also drawn from WebArena, where GPT-4o agent with LCoW succeeded, while GPT-4o with raw observation failed. In this example, the contextualization module explains that the agent need to click "Profile" link in order to edit posts or submission, thereby leading to correct decision making of LLM agent. However, given raw observation, the GPT-4o agent consecutively scrolls down the page to find the corresponding post.

- **Example 4**. In this example, the contextualized observation contains an explanation about the UI for keyword search, thereby guiding the GPT-4o agent to find a white desk based on search functionality. However, the GPT-4o agent without LCoW repeatedly scrolls down the page and moves on to the next page to find the white desk, thereby failing to add a white desk to the wish list within the maximum step limit.

- **Example 5**. In the task of filling out a form to create a new user, it is crucial to check for detailed requirements and entries before submitting the form. LCoW- contextualized web page includes, thereby guiding LLM agents to submit the complete form without mistakes (e.g., missing required entries).

---

**Example 1 (WorkArena)**

**Instruction:** Sort the "users" list by the following fields:- Updates (descending)

**Contextualized web page observation:**
```
[a183] grid 'Users.', multiselectable=False
This is the main table containing the list of users.
We need to focus on this element to find ways to interact
with the column header.
[a186] rowgroup ''
  [a187] row ''
    [a188] columnheader 'Select All'
    [a193] columnheader '\uf1e4 Show column search row'
    [a197] columnheader 'User ID User ID column options'
    [a204] columnheader 'Name Name column options'
    [a211] columnheader 'Email Email column options'
    [a218] columnheader 'Avatar Avatar column options'
    [a225] columnheader 'Title Title column options'
```

```
    [a232] columnheader 'Company Company column options'
    [a239] columnheader 'Department Department column options'
    [a246] columnheader 'Location Location column options'
    [a253] columnheader 'Time zone Time zone column options'
    [a260] columnheader 'Business phone Business phone column options'
These are the column headers in the users table.
We need to focus on one of these elements to trigger
the sorting action.
However, the AXTree does not show any direct options to
sort these columns.

[a77] button '\uf13e Personalize List'
This button might provide options to customize
the list view, potentially including sorting options.
```

### Example 2 (WebArena)

**Instruction:** Tell me the status of my latest order and when will it arrive.

**Contextualized web page observation:**

```
[1415] strong ''
    StaticText 'Recent Orders'
[1416] link 'View All'
[1420] table 'Recent Orders'
    [1421] caption ''
    [1422] rowgroup ''
        [1423] row ''
            [1424] columnheader 'Order #'
            [1425] columnheader 'Date'
            [1426] columnheader 'Ship To'
            [1427] columnheader 'Order Total'
            [1428] columnheader 'Status'
            [1429] columnheader 'Action'
    [1430] rowgroup ''
        [1431] row ''
            [1432] gridcell '000000170'
            [1433] gridcell '5/17/23'
            [1434] gridcell 'Emma Lopez'
            [1435] gridcell '$365.42'
            [1437] gridcell 'Canceled'
            [1438] gridcell 'View OrderReorder'
                [1439] link 'View Order'
                [1441] link 'Reorder'

This extraction shows the "Recent Orders" section,
which contains a table with information about recent orders.
The table includes columns for Order #, Date, Ship To,
Order Total, Status, and Action.
The "View All" link [1416] likely leads to a page with more
detailed information about all orders.

The first row in the table [1431] contains information about
the latest order:
- Order number: 000000170
- Date: 5/17/23
- Ship To: Emma Lopez
- Order Total: $365.42
- Status: Canceled
- Action: View Order [1439] or Reorder [1441]
```

**Example 3 (WebArena)**

**Instruction:** Edit my post on Star Trek Starfleet Academy series by adding a line to the body that says
"Every watch makes me feel like a kid again.

**Contextualized web page observation:**

```
[68] button 'MarvelsGrantMan136', focused,
              hasPopup='menu', expanded=True
    This is the currently focused and expanded user
    account menu button.
    It contains the user's username and is likely to
    lead to profile settings.

[74] list ''
    This list contains the submenu items under
    the user account menu.

[80] listitem ''
    [81] link 'Profile'
        This is the "Profile" link we need to click to
        access the user's profile settings.
        It's nested under the user account menu.

These elements are crucial for navigating to the user's
profile settings, where we can then look for options to edit
posts or submissions.
The "Profile" link [81] is particularly important as it's the
next element we need to interact with to progress towards
our goal of editing the post.
```

**Example 4 (WebArena)**

**Instruction:** Add a white desk to my wish list.

**Contextualized web page observation:**

```
[1850] heading 'My Wish List'
StaticText 'You have no items in your wish list.'

This element shows the "My Wish List" heading and
indicates that the wish list is currently empty.
It's important to note this because it confirms
that we haven't added any items to the wish list yet,
including the white desk we want to add.

[1430] button 'Add to Wish List'
    StaticText '\ue601'

This button is crucial as it directly allows us to
add items to the wish list.
The button with ID 1430 is likely the one we need
to click to add the white desk to the wish list.

[723] menuitem '\ue622 Office Products', hasPopup='menu'

This menuitem is relevant because it's the currently
selected category (Office Products).
It confirms that we are in the correct category
to find office-related items, including desks.

[269] Section ''
    [271] LabelText ''
        StaticText '\ue615'
        StaticText 'Search'
```

```
    [274] combobox '\ue615 Search',
        autocomplete='both', hasPopup='listbox', expanded=False
    [277] link 'Advanced Search'
    [279] button 'Search', disabled=True

This section contains the search functionality,
which could be useful if we need to search for "white desk"
to find the specific item we want to add to the wish list.

These elements provide the necessary information and
functionality to complete the user's instruction of
adding a white desk to the wish list.
```

**Example 5 (WorkArena)**

**Instruction:** Create a new user with a value of "8273378363827782819" for field "User ID", a value of "Webster-Richard" for field "Last name", a value of "Lynn-David" for field "First name", a value of "lynn-david.webster-richard@workarena.com" for field "Email", a value of "false" for field "Internal Integration User", a value of "Outlook" for field "Calendar integration", a value of "(514) 357-8997" for field "Mobile phone", a value of "false" for field "Password needs reset", a value of "English" for field "Language", and a value of "false" for field "Web service access only".

**Contextualized web page observation:**

```
we should focus on the following elements:

[a298] LabelText ''
    [a299] note ''
    StaticText 'Internal Integration User'
[a305] checkbox 'Internal Integration User', checked='false'

This element represents the "Internal Integration User" field.
It's a checkbox that is currently unchecked (false), which
aligns with the user instruction.
No action is needed for this field.

[a353] LabelText ''
    [a354] note ''
    StaticText 'Calendar integration'
[a359] combobox 'Calendar integration' value='Outlook',
hasPopup='menu', expanded=False
    [a360] menuitem '-- None --', selected=False
    [a361] menuitem 'Outlook', selected=True

This element represents the "Calendar integration" field.
It's a combobox that is already set to "Outlook", which
aligns with the user instruction.
No action is needed for this field.

[a429] LabelText ''
    [a430] note ''
    StaticText 'Mobile phone'
[a435] textbox 'Mobile phone'

This element represents the "Mobile phone" field.
It's a textbox where we need to enter "(514) 357-8997" as per
the user instruction.

[a327] LabelText ''
    [a328] note ''
    StaticText 'Language'
[a333] combobox 'Language' value='-- None --',
                        hasPopup='menu', expanded=False
    [a334] menuitem '-- None --', selected=True
```

```
    [a335] menuitem 'English', selected=False

This element represents the "Language" field.
It's a combobox where we need to select "English" as per the
user instruction.

[a285] LabelText ''
    [a286] note ''
    StaticText 'Web service access only'
[a292] checkbox 'Web service access only', checked='false'
```

## A.2 ANALYSIS ON DECISION MAKING PROCESS

We provide the action sequences obtained from rollouts in evaluation tasks. For a given instruction, we compare the rollouts with and without the application of LCoW, highlighting the differences in decision-making behavior and task completion efficiency. Specifically, LCoW helps the LLM agent to accomplish task more efficiently without repeating redundant actions (see Figure 10 and Figure 12), and also helps to avoid getting stuck by informing proper interaction method regarding UI elements (see Figure 11 and Figure 13). Furthermore, as shown in Figure 14, with the contextualization module, the average number of decision-making steps by the agent decreases in both Claude-3.5-Sonnet and Llama-3.1-70B agent, allowing tasks to be completed more efficiently.

**Instruction:** Go to the hardware store and order 9 "iPad pro" with configuration {'Choose the colour': 'Silver', 'Choose the storage': '128'}

| Llama-3.1-70B | Llama-3.1-70B + LCoW |
|---|---|
| **Action 1**: click [a252] link 'Hardware' | **Action 1**: click [a252] link 'Hardware' |
| **Action 2**: click [a268] link 'iPad pro' | **Action 2**: click [a268] link 'iPad pro' |
| **Action 3**: select 'Silver' from [a198] radiogroup 'Mandatory' | **Action 3**: click [a184] radio '\uf137 Silver' |
| **Action 4**: click [a184] radio '\uf137 Silver' | **Action 4**: click [a185] LabelText '' |
| **Action 5**: hover the cursor on [a184] radio '\uf137 Silver' | **Action 5**: select '9' from [a244] combobox 'Quantity' |
| **Action 6**: click [a185] LabelText '' | **Action 6**: click [a285] button 'Order Now' |
| **Action 7**: select '9' from [a244] combobox 'Quantity' | |
| **Action 8**: click [a285] button 'Order Now' | |

Figure 10: Llama-3.1-70B agent repeats redundant actions while manipulating UI element for selecting silver color, while LCoW helps to decide on actions more efficiently.

**Instruction:** Add a white desk to my wish list.

| GPT-4o | GPT-4o + LCoW |
|---|---|
| **Action 1**: click [716] menuitem '\ue622 Office Products' | **Action 1**: click [716] menuitem '\ue622 Office Products' |
| **Action 2**: click [1809] link 'Office Furniture&Lighting(372 item)' | **Action 2**: fill 'white desk' in [274] combobox '\ue615 Search' |
| **Action 3**: click [1789] link '\ue608 Page Next' | **Action 3**: click [279] button 'Search' |
| **Action 4**: click [1815] link '\ue608 Page Next' | **Action 4**: click [1434] button 'Add to Wish List' |
| **Action 5**: scroll_down | |
| **Action 6**: scroll_down | |
| **Action 7**: scroll_down | |

Figure 11: Given complex web page observation, GPT-4o agent repeats moving to the next page and scrolling down the page for finding out white desk, resulting in task failure. In contrast, LCoW enables LLMs to utilize search functionality for finding out the white desk efficiently.

**Instruction:** I recently moved, my address is 111 Magnolia Path, Atlanta, GA, 30303, update my information on OneStopShopping accordingly.

| GPT-4o | GPT-4o + LCoW |
|---|---|
| **Action 1**: click [181] link 'My Account' | **Action 1**: click [219] link 'My Account' |
| **Action 2**: click [1398] link 'Edit Address' | **Action 2**: click [1384] link 'Manage Addresses' |
| **Action 3**: clear contents in [1400] textbox 'Street Address* Street Address: Line 1' | **Action 3**: click [1374] link 'Change Billing Address' |
| **Action 4**: fill '111 Magnolia Path' in [1400] textbox 'Street Address* Street Address: Line 1' | **Action 4**: fill '111 Magnolia Path' in [1400] textbox 'Street Address* Street Address: Line 1' |
| **Action 5**: clear contents in [1737] textbox 'City *' | **Action 5**: fill 'Atlanta' in [1737] textbox 'City *' |
| **Action 6**: fill 'Atlanta' in [1737] textbox 'City *' | **Action 6**: select 'Georgia' from [1665] combobox 'State/Province *' |
| **Action 7**: select 'Georgia' from [1665] combobox 'State/Province *' | **Action 7**: fill '30303' in [1742] textbox 'Zip/Postal Code *' |
| **Action 8**: clear contents in [1742] textbox 'Zip/Postal Code *' | **Action 8**: click [1752] button 'Save Address' |
| **Action 9**: fill '30303' in [1742] textbox 'Zip/Postal Code *' | |
| **Action 10**: click [1752] button 'Save Address' | |

Figure 12: GPT-4o agent meaninglessly clear the empty textbox before filling out the address form. In contrast, LCoW explains the status of UI element (e.g., textbox is empty), thereby enabling more efficient decision making.

**Instruction:** Post a notice on a virtual meetup for Harry Potter enthusiasts on July 8th in the books subreddit.

| GPT-4o | GPT-4o + LCoW |
|---|---|
| **Action 1**: click [63] link 'Submit' | **Action 1**: click [62] link 'Submit' |
| **Action 2**: fill 'Virtual Meetup for Harry Potter Enthusiasts on July 8th' in [137] textbox 'Title This field is required.' | **Action 2**: fill 'Join Our Harry Potter Virtual Meetup on July 8th!' in [137] textbox 'Title This field is required.' |
| **Action 3**: fill 'Join us for a virtual meetup for Harry Potter enthusiasts on July 8th! We will discuss our favorite books in [142] textbox 'Body' | **Action 3**: fill 'Join us for a magical virtual meetup where Harry Potter enthusiasts from around the world come together to discuss their favorite books in [142] textbox 'Body' |
| **Action 4**: click [378] combobox 'Choose one…' | **Action 4**: fill 'books' in [379] textbox 'Choose one…' |
| **Action 5**: select 'books' from [378] combobox 'Choose one…' | **Action 5**: click [378] combobox 'Choose one…' |
| **Action 6**: press 'ArrowDown' keys while focusing on [378] combobox 'Choose one…' | **Action 6**: click [453] option 'books' |
| **Action 7**: press 'ArrowDown' keys while focusing on [378] combobox 'Choose one…' | **Action 7**: click [395] button 'Create submission' |
| **Action 8**: press 'ArrowDown' keys while focusing on [378] combobox 'Choose one…' | |

Figure 13: GPT-4o agent struggles to interact with UI element for selecting subreddit, resulting in task failure. In contrast, LCoW verbally explains the interaction method of each UI element in the web page, enabling task success.

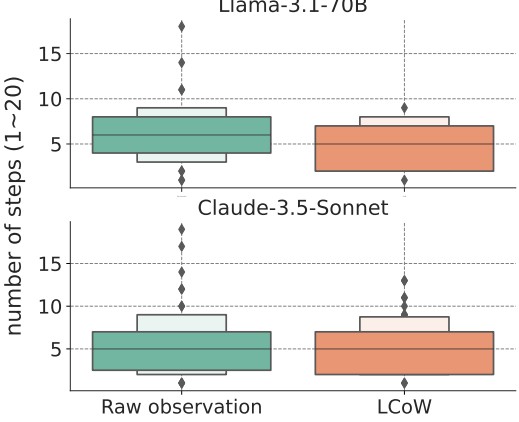

Figure 14: In order to compare the efficiency of task execution between LLM agent with LCoW and without LCoW, we select the tasks that LLM agent with LCoW and without LCoW both succeed, and visualize the distribution of action steps required for accomplishing the task.

# B  EXPERIMENTAL DETAILS & ADDITIONAL EXPERIMENTS

## B.1  MAIN EXPERIMENT SETUP

**WebShop**  For the WebShop benchmark, we fine-tune Phi-3-mini-Instruct as a contextualization module, setting the learning rate, warmup ratio, and batch size to 1e-5, 1e-2, and 32, respectively. The module is trained for a single epoch over the collected data for each LCoW iteration, and we utilize demonstrations corresponding to 397 individual tasks provided in the WebShop benchmark as seed demonstrations. In the WebShop environment, we employ Gemini-1.5-flash as the LLM agent, combined with the contextualization module during the trajectory collection phase. It is also utilized for sampling optimal contextualization at the initial iteration of LCoW.

**WorkArena & WebArena**  For WorkArena and WebArena, we fine-tune Llama-3.1-8B-Instrcut as an observation contextualization module, and we set the learning rate, warm-up ratio, and batch size as 1e-5, 1e-1, and 128. Additionally, we set the training epochs as 4 and 3 for WorkArena and WebArena, respectively. Since no successful trajectories are provided in WorkArena, we collected 264 seed demonstrations (i.e., successful trajectories) among 495 training tasks across 33 task types using Claude-3.5-Sonnet and GPT-4o, but we could not collect any successful trajectories corresponding to 10 task types. Summary statistics of the collected seed demonstrations across task types can be found in Table 5. In WebArena, we collected 363 seed demonstrations (i.e., successful trajectories) among 647 training tasks (excluding 165 evaluation tasks in WebArena-Lite from 812 WebArena tasks) using GPT-4o and trajectories open-sourced by AgentOccam (Yang et al., 2024). Additionally, as the determination of action matching based on parsing is infeasible due to open-ended actions (e.g., sending a message to the user or filling out a form), we exploit GPT-4o as an action-matching evaluator. The prompt used for the action-matching evaluator is provided in Appendix C. For WorkArena and WebArena, we use Claude-3.5-Sonnet for sampling optimal contextualization at the initial iteration.

## B.2  GENERALIZATION EXPERIMENT SETUP

In this section, we first provide details of the generalization experiment in WebArena and WorkArena.

**WebArena**  In WebArena, we define unseen-type tasks based on the task template used to create 812 individual tasks in the WebArena benchmark. Specifically, 812 tasks in WebArena are all different but created based on 190 task templates. Therefore, tasks created from the same task template are different but similar. For example, "What is the top-1 best-selling product in 2022" and "What is the top-3 best-selling product in 2023" are from the same task template (i.e., same task type), while "Tell me the email address, name, phone number of the customer who has the most cancellations in the history" and "How many reviews our shop received in Apr 2023?" are from the different task template (i.e., different task type). Among the 165 evaluation tasks, we define the 117 tasks whose task templates overlap with tasks used for training as seen-type tasks, while the 48 tasks whose task templates do not overlap with any training tasks as unseen-type tasks.

**WorkArena**  WorkArena features a two-level task hierarchy: task categories at the top level, and task types within each category. WorkArena provides 7 categories of tasks (i.e., "Dashboard", "Menus", "Service catalog", "Knowledge base", "Forms", "Sort list", and "Filter list"). For instance, "Form" is a task *category*, and within it, "creating and submitting an incident report" and "creating new user information" are task *types*. We consider two levels of generalization: 1) Unseen-type tasks, tasks of a different type within the same category (i.e., medium-level generalization), and 2) Unseen-category tasks, tasks of a different type and category (i.e., hard-level generalization). As specified in Table 6, among 33 task types, we utilize 13 task types as seen task types, 14 task types as unseen task types, and the remaining 6 task types corresponding to "Filter List" task category as unseen-category tasks. For evaluation, we utilize 5 individual tasks from each task type.

### B.3 Additional generalization experiment results

In this section, we provide the result of additional generalization experiments in the WorkArena and WebArena benchmarks.

**Generalization to tasks with unseen type in WorkArena**  We first evaluate generalization to unseen-type tasks in the WorkArena benchmark, where 70 tasks corresponding to 14 task types are considered unseen-type tasks. As shown in Table 3, LCoW improves GPT-4o and Gemini-1.5-flash agents by 7.2% and 22.6% success rate in 70 unseen-type tasks, respectively. This is an observation consistent with the WebArena experiment in Section 4.3.

**Generalization to tasks with unseen category in WorkArena**  Furthermore, we verify whether the contextualization module generalizes to tasks from entirely different categories (i.e., unseen-category tasks) in the WorkArena benchmark. This requires an understanding of unseen UI elements and webpage configurations, making it more challenging than generalization to unseen-type tasks. Specifically, we set the 30 tasks included in "`Filter-List`" task category as unseen-category tasks. As shown in Table 3, LCoW fails to improve both GPT-4o and Gemini-1.5-flash agents in unseen-category tasks. The main reason for this failure in the unseen-category tasks is that the contextualization module does not extract the necessary UI element required to dropdown the hidden menu when manipulating filters in `Filter-List`-related tasks. Since the training tasks do not involve any UI elements related to filter functionality, the contextualization module does not learn any knowledge about the UI element during training, resulting in no performance improvement in unseen-category tasks.

|                  | GPT-4o      |                  | Gemini-1.5-flash |                  |
| ---------------- | ----------- | ---------------- | ---------------- | ---------------- |
|                  | Unseen-type | Unseen-category  | Unseen-type      | Unseen-category  |
| Raw observation  | 35.7%       | 0.0%             | 14.5%            | 0.0%             |
| LCoW             | **42.9%**   | 0.0%             | **37.1%**        | 0.0%             |

Table 3: LCoW is shown to be generalized to unseen task types within the same task category on both GPT-4o and Gemini-1.5-flash backbone, but it struggles to generalize to tasks corresponding to unseen category.

**Generalization to tasks with unseen websites**  We additionally study the feasibility of LCoW for generalization to unseen websites in the WebArena benchmark, where tasks are distributed over 6 websites (Shopping, GitLab, Reddit, Map, Content Management System, and Wikipedia). We evaluate the contextualization module on tasks defined in the Shopping website, where the contextualization module is trained based on tasks in the remaining 5 websites. In this setting, we found that LCoW improves the performance of the GPT-4o agent by 4.3% even in the unseen website, as shown in Table 4. We conjecture that this is feasible because, even though the websites may differ in overall design and layout, common UI elements (e.g., search query input fields and calendar date selectors) are often shared across websites. The presence of shared UI components allows the knowledge gained from one website's task to be transferred effectively to another website, enabling the contextualization to generalize to tasks in unseen websites. We believe scaling LCoW for generalization to unseen websites where unseen UI elements are prevalent is a highly interesting future direction.

| (Unseen website) | GPT-4o    |
| ---------------- | --------- |
| Raw observation  | 17.4%     |
| LCoW             | **21.7%** |

Table 4: LCoW improves the performance of GPT-4o agent on tasks corresponding to the unseen website. Although websites are different, there are UI elements commonly used across websites (e.g., UI elements for searching in Map or UI elements for filtering results based on data in the content management system), which enables LCoW to generalize across different websites.

| Task type | Number of seed demonstrations |
|---|:---:|
| Multi-chart-value retrieval | 8 |
| Multi-chart-minmax retrieval | 12 |
| Single-chart-value-retrieval | 12 |
| Single-chart-minmax retrieval | 11 |
| Create change request | 8 |
| Create incident | 0 |
| Create hardward asset | 0 |
| Create problem | 14 |
| Create user | 11 |
| Knowledge base search | 12 |
| Filter asset list | 0 |
| Filter change request list | 0 |
| Filter hardware list | 0 |
| Filter incident list | 0 |
| Filter service catalog item list | 0 |
| Filter user list | 0 |
| Sort asset list | 0 |
| Sort change request list | 0 |
| Sort hardware list | 3 |
| Sort incident list | 3 |
| Sort service-catalog list | 5 |
| Sort user list | 4 |
| All menu | 15 |
| Impersonation | 12 |
| Order developer laptop | 15 |
| Order iPad mini | 14 |
| Order iPad pro | 15 |
| Order sales laptop | 15 |
| Order standard laptop | 15 |
| Order apple watch | 15 |
| Order apple Macbook pro | 15 |
| Order development laptop PC | 15 |
| Order loander laptop | 15 |
| Total | 264 |

Table 5: We collected trajectories from 15 individual tasks for each of 33 task types in the WorkArena benchmark using GPT-4o-0806 and Claude-3.5-sonnet agent, thereby collecting 264 successful trajectories. We utilized them as seed demonstrations for LCoW.

| Task type | Seen / Unseen |
|---|---|
| Single-chart-value-retrieval | seen |
| Single-chart-minmax retrieval | seen |
| Multi-chart-value retrieval | unseen-type |
| Multi-chart-minmax retrieval | unseen-type |
| Create change request | seen |
| Create problem | seen |
| Create incident | unseen-type |
| Create hardward asset | unseen-type |
| Create user | unseen-type |
| Knowledge base search | seen |
| Sort user list | seen |
| Sort hardware list | seen |
| Sort asset list | unseen-type |
| Sort change request list | unseen-type |
| Sort incident list | unseen-type |
| Sort service-catalog list | unseen-type |
| All menu | seen |
| Impersonation | unseen-type |
| Order developer laptop | seen |
| Order iPad mini | seen |
| Order iPad pro | seen |
| Order apple watch | seen |
| Order standard laptop | seen |
| Order sales laptop | unseen-type |
| Order apple Macbook pro | unseen-type |
| Order development laptop PC | unseen-type |
| Order loander laptop | unseen-type |
| Filter asset list | unseen-category |
| Filter change request list | unseen-category |
| Filter hardware list | unseen-category |
| Filter incident list | unseen-category |
| Filter service catalog item list | unseen-category |
| Filter user list | unseen-category |

Table 6: Task type split for evaluation of generalization at different levels in WorkArena benchmark.

# C PROMPTS

In this section, we provide prompts used across experiments in WebShop, WorkArena, and WebArena. In WorkArena and WebArena experiments, we utilize identical prompts, where the prompt for the LLM agent is provided by BrowserGym (Drouin et al., 2024). In the WebShop experiment, we utilize the prompt for the LLM agent proposed by ReAct (Yao et al., 2022b).

## C.1 WORKARENA & WEBARENA

We present the prompt formats for the experiment in the WorkArena and WebArena benchmark.

---

**Prompt for contextualization module**

```
<system prompt>
You are an agent tasked with extracting and refining a subset of the
webpage's observations based on the content of the page and user
instructions.

<main prompt>
You are currently on the {domain_info} website.
Your task is to generate a "Reasoning" and a "Refined observation"
based on the provided inputs.

First, review the "User instruction" and "History of interactions"
and, then, generate the "Reasoning".
Analyze the progress made so far, and provide a rationale for the
next steps needed to efficiently accomplish the user instruction on
the {domain_info} website.

Second, refine the "AXTree observation at the current time step"
into a "Refined observation".
Select a subset of the AXTree observation that is essential for
completing the user instruction and provide explanations for the
corresponding elements in the selected subset.

[Information source]
# User instruction
{goal}

# History of interactions
{history}

# AXTree observation at the current time step
{observation}
```

---

**Prompt for contextualization module in retry phase**

```
<system prompt>
You are an agent tasked with extracting and refining a subset of the
webpage's observations based on the content of the page and user
instructions.

<main prompt>
You are currently on the {domain_info} website.
Your task is to generate a "Reasoning" and a "Refined observation"
based on the provided inputs.

First, review the "User instruction" and "History of interactions"
and, then, generate the "Reasoning".
Analyze the progress made so far, and provide a rationale for the
next steps needed to efficiently accomplish the user instruction on
the {domain_info} website.
```

---

```
Second, refine the "AXTree observation at the current time step"
into a "Refined observation".
Select a subset of the AXTree observation that is necessary for
completing the user instruction.

You may refer to the Hints, which consists of the ground truth next
action, but do not explicitly mention these hints in your output.

[Information source]
# User instruction
{goal}

# History of interactions
{history}

# AXTree observation at the current time step
{observation}

# Hint
Ground-truth next action: {action}
```

**Prompt for self-contextualization**

```
<system prompt>
You are an agent tasked with extracting and refining a subset of the
webpage's observations based on the content of the page and user
instructions.

<main prompt>
[General instructions]
You are currently on the {domain_info} website.
Your task is to generate a "Reasoning" and a "Refined observation"
based on the provided inputs.

First, review the "User instruction" and "History of interactions" and,
then, generate the "Reasoning".
Analyze the progress made so far, and provide a rationale for the next
steps needed to efficiently accomplish the user instruction on
the {domain_info} website.

Second, refine the "AXTree observation at the current time step"
into a "Refined observation".
Extract a subset of the AXTree observation (e.g., chart, table,
menu items) that contains necessary information for completing
the user instruction, and explain the extracted elements.
Ensure that the information on the elements (e.g., numeric element ID)
are correctly included.

Please follow the format in the [Reasoning & Refinement example]
carefully.

[Information source]
# User instruction
{goal}

# History of interactions
{history}

# AXTree observation at the current time step
{observation}

[Reasoning & Refinement example]
# Abstract example
```

```
Here is an abstract version of the answer, describing
the content of each tag.
Make sure you follow this structure, but replace the
content with your own answer:

<reasoning>
Think step by step. Based on the "User instruction,",
"History of interaction," and "AXTree observation at the
current time step":
1. Provide a high-level description of the "AXTree observation at the
current time step."
2. Based on the "User instruction" and "History of interaction,"
track your progress and provide your reasoning on the next action
needed to accomplish the "User instruction."
</reasoning>

<extraction>
Based on your reasoning, identify the elements
(e.g., links, buttons, static text, table row, chart) to focus on.
Then, explain the semantics and functionalities of
each extracted element.
Ensure that:
You do not alter the structure of the AXTree observation.
You extract the element ID (id in [ ]) accurately without any errors.
When extracting chart or table, you must extract the entire chart
or table to avoid any confusion or loss of information.
</extraction>
```

**Prompt for LLM agent**

```
<system prompt>
You are an agent trying to solve a web task based on the content of
the page and a user instructions.
You can interact with the page and explore.
Each time you submit an action it will be sent to the browser and you
will receive a new page.

<main prompt>
# Instructions
Review the current state of the page and all other information to find
the best possible next action to accomplish your goal.
Your answer will be interpreted and executed by a program, make sure
to follow the formatting instructions.

## Goal:
{goal}

{history}

# Refined observation of current step:
{refined observation}

# Action space:
13 different types of actions are available.
noop(wait_ms: float = 1000)
    Description: Do nothing, and optionally wait for
    the given time (in milliseconds).
    Examples:
        noop()
        noop(500)

send_msg_to_user(text: str)
    Description: Send a message to the user.
    You should send a short answer as a message and
```

```
    do not ask questions through message.
    Examples:
        send_msg_to_user(\'the city was built in 1751.\')
        send_msg_to_user(\'Yes\')
        send_msg_to_user(\'No\')
        send_msg_to_user(\'31112\')
        send_msg_to_user(\'Yoshua Bengio\')

scroll(delta_x: float, delta_y: float)
    Description: Scroll horizontally and vertically.
    Amounts in pixels, positive for right or down scrolling,
    negative for left or up scrolling. Dispatches a wheel event.
    Examples:
        scroll(0, 200)
        scroll(-50.2, -100.5)

fill(bid: str, value: str)
    Description: Fill out a form field.
    It focuses the element and triggers an input event
    with the entered text. It works for <input>,
    <textarea> and [contenteditable] elements.
    Examples:
        fill('237', 'example value')
        fill('45', 'multi-line\nexample')
        fill('a12', 'example with "quotes"')

select_option(bid: str, options: str | list[str])
    Description: Select one or multiple options in a <select> element.
    You can specify option value or label to select.
    Multiple options can be selected.
    Examples:
        select_option('48', 'blue')
        select_option('48', ['red', 'green', 'blue'])

click(bid: str, button: Literal['left', 'middle', 'right'] = 'left',
    modifiers: list[typing.Literal['Alt', 'Control', 'Meta', 'Shift']]
    = [])
    Description: Click an element.
    Examples:
        click('51')
        click('b22', button='right')
        click('48', button='middle', modifiers=['Shift'])

dblclick(bid: str, button: Literal['left', 'middle', 'right'] =
'left', modifiers: list[typing.Literal['Alt', 'Control', 'Meta',
'Shift']] = [])
    Description: Double click an element.
    Examples:
        dblclick('12')
        dblclick('ca42', button='right')
        dblclick('178', button='middle', modifiers=['Shift'])

hover(bid: str)
    Description: Hover over an element.
    Examples:
        hover('b8')

press(bid: str, key_comb: str)
    Description: Focus the matching element and press a combination of
    keys.
    It accepts the logical key names that are emitted in the
    keyboardEvent.
    key property of the keyboard events: Backquote, Minus, Equal,
```

```
        Backslash, Backspace, Tab, Delete, Escape, ArrowDown, End, Enter,
        Home, Insert, PageDown, PageUp, ArrowRight, ArrowUp, F1 - F12,
        Digit0 - Digit9, KeyA - KeyZ, etc. You can alternatively specify a
        single character you'd like to produce such as "a" or "#".
        Following modification shortcuts are also supported: Shift,
        Control, Alt, Meta.
        Examples:
            press('88', 'Backspace')
            press('a26', 'Control+a')
            press('a61', 'Meta+Shift+t')

focus(bid: str)
    Description: Focus the matching element.
    Examples:
        focus('b455')

clear(bid: str)
    Description: Clear the input field.
    Examples:
        clear('996')

drag_and_drop(from_bid: str, to_bid: str)
    Description: Perform a drag & drop.
    Hover the element that will be dragged.
    Press left mouse button.
    Move mouse to the element that will receive the drop.
    Release left mouse button.
    Examples:
        drag_and_drop('56', '498')

upload_file(bid: str, file: str | list[str])
    Description: Click an element and wait for a "filechooser" event,
    then select one or multiple input files for upload.
    Relative file paths are resolved relative to the current working
    directory.
    An empty list clears the selected files.
    Examples:
        upload_file('572', 'my_receipt.pdf')
        upload_file('63', ['/home/bob/Documents/image.jpg',
        '/home/bob/Documents/file.zip'])

Only a single action can be provided at once. Example:
fill('a12', 'example with "quotes"')
Multiple actions are meant to be executed sequentially without any
feedback from the page.
Don't execute multiple actions at once if you need feedback from the
page.

# Abstract Example
Here is an abstract version of the answer with description of the
content of each tag.
Make sure you follow this structure, but replace the content with your
answer:

<think>
Think step by step.
If you need to make calculations such as coordinates, write them here.
Describe the effect that your previous action had on the current
content of the page.
</think>

<action>
One single action to be executed.
You can only use one action at a time.
```

```
</action>

# Concrete Example
Here is a concrete example of how to format your answer.
Make sure to follow the template with proper tags:

<think>
My memory says that I filled the first name and last name, but I can't
see any content in the form.
I need to explore different ways to fill the form.
Perhaps the form is not visible yet or some fields are disabled.
I need to replan.
</think>

<action>
fill('a12', 'example with "quotes"')
</action>
```

## C.2 WEBSHOP

We present the prompt formats for the experiment in the WebShop benchmark.

**Prompt for contextualization module**

```
<system prompt>
You are an agent tasked with extracting and rephrasing a subset of
the webpage's observations based on the content of the page and user
instructions.

<main prompt>
You are currently on the online shopping website.
Your task is to generate a "Reasoning" and a "Refined observation"
based on the provided inputs.

First, review the "User instruction" and "History of interactions"
and, then, generate the "Reasoning".
Analyze the progress made so far, and provide a rationale for the
next steps needed to efficiently accomplish the user instruction on
the online shopping website.

Second, rephrase the "AXTree observation at the current time step"
into a "Rephrased observation".
Select a subset of the AXTree observation that is essential
for completing the user instruction and provide explanations
for the corresponding elements in the selected subset.

[Information source]
# User instruction
{goal}

# History of interactions
{previous_actions}

# AXTree observation at the current time step
{obs}
```

**Prompt for self-contextualization**

```
<system prompt>
You are an agent tasked with extracting and rephrasing a subset of
the webpage's observations based on the content of the page and user
instructions.
```

```
<main prompt>
The current webpage on the web shopping site is described
in the observation.
Evaluate the current progress based on previous actions
and current observation.
Determine the next action by reasoning based on
goal and progress.
Condense the observation into a concise format, highlighting
clickable buttons indicated by [].
Ensure the summary includes only elements relevant to the
goal and not already covered in previous actions.
Focus on clickable buttons indicated as [].

Here are a few examples.

**goal**: i would like a 3 ounce bottle of bright citrus
deodorant for sensitive skin, and price lower than 50.00 dollars
**previous actions**:
1. search[3 ounce bright citrus deodorant sensitive skin]
**current observation**:
[ Back to Search ]
Page 1 (Total results: 50)
[ Next > ]
[ B078GWRC1J ]
Bright Citrus Deodorant by Earth Mama | Natural and Safe
for Sensitive Skin, Pregnancy and Breastfeeding,
Contains Organic Calendula 3-Ounce
$10.99
[ B078GTKVXY ]
Ginger Fresh Deodorant by Earth Mama | Natural and Safe for
Sensitive Skin, Pregnancy and Breastfeeding, Contains Organic
Calendula 3-Ounce
$10.99
[ B08KBVJ4XN ]
Barrel and Oak - Aluminum-Free Deodorant, Deodorant for Men,
Essential Oil-Based Scent, 24-Hour Odor Protection, Cedar &
Patchouli Blend, Gentle on Sensitive Skin (Mountain Sage,
2.7 oz, 2-Pack)
$15.95

**rephrased observation**:
Progress: I searched the keyword '3 ounce bright citrus deodorant
sensitive skin' to see the relvant items, And now I am looking at
the item list.
Reasoning: the next step is to choose an item satisfying the
specification of bright citrus deodorant.
I can focus on:
[B078GWRC1J]
Bright Citrus Deodorant by Earth Mama | Natural and Safe for
Sensitive Skin, Pregnancy and Breastfeeding, Contains Organic
Calendula 3-Ounce
$10.99

**goal**: i would like a 3 ounce bottle of bright citrus deodorant
for sensitive skin, and price lower than 50.00 dollars
**previous actions**:
1. search[3 ounce bright citrus deodorant sensitive skin]
2. click[B078GWRC1J]
**current observation**:
[ Back to Search ]
[ < Prev ]
size
```

```
[ travel set (4-pack) ]
[ 3 ounce (pack of 1) ]
[ 3-ounce (2-pack) ]
scent
[ assorted scents ]
[ bright citrus ]
[ calming lavender ]
[ ginger fresh ]
[ simply non-scents ]
Bright Citrus Deodorant by Earth Mama | Natural and Safe for
Sensitive Skin, Pregnancy and Breastfeeding, Contains Organic
Calendula 3-Ounce
Price: $10.99
Rating: N.A.
[ Description ]
[ Features ]
[ Reviews ]
[ Buy Now ]

**rephrased observation**:
Progress: I searched and and clicked the item seems to be most
relevant to the goal specification. I am looking at the option list.
Reasoning: As the goal requires 3-ounce bottle, I can focus
on the size option.
I can focus on:
size
[ travel set (4-pack) ]
[ 3 ounce (pack of 1) ]
[ 3-ounce (2-pack) ]

**goal**: i would like a 3 ounce bottle of bright citrus deodorant
for sensitive skin, and price lower than 50.00 dollars
**previous actions**:
1. search[3 ounce bright citrus deodorant sensitive skin]
2. click[B078GWRC1J]
3. click[3 ounce (pack of 1)]
**current observation**:
You have clicked 3 ounce (pack of 1).
[ Back to Search ]
[ < Prev ]
size
[ travel set (4-pack) ]
[ 3 ounce (pack of 1) ]
[ 3-ounce (2-pack) ]
scent
[ assorted scents ]
[ bright citrus ]
[ calming lavender ]
[ ginger fresh ]
[ simply non-scents ]
Bright Citrus Deodorant by Earth Mama | Natural and Safe for
Sensitive Skin, Pregnancy and Breastfeeding, Contains Organic
Calendula 3-Ounce
Price: $10.99
Rating: N.A.
[ Description ]
[ Features ]
[ Reviews ]
[ Buy Now ]

**rephrased observation**:
Progress: I searched and and clicked the item id.
Among the option list, and I clicked size option.
```

```
Reasoning: According to the progress, I have to focus
on the scent option as a next step.
I can focus on:
scent
[ assorted scents ]
[ bright citrus ]
[ calming lavender ]
[ ginger fresh ]
[ simply non-scents ]

**goal**: i would like a 3 ounce bottle of bright citrus
deodorant for sensitive skin, and price lower than 50.00 dollars
**previous actions**:
1. search[3 ounce bright citrus deodorant sensitive skin]
2. click[B078GWRC1J]
3. click[3 ounce (pack of 1)]
4. click[bright citrus]
**current observation**:
You have clicked 3 ounce (pack of 1).
You have clicked bright citrus.
[ Back to Search ]
[ < Prev ]
size
[ travel set (4-pack) ]
[ 3 ounce (pack of 1) ]
[ 3-ounce (2-pack) ]
scent
[ assorted scents ]
[ bright citrus ]
[ calming lavender ]
[ ginger fresh ]
[ simply non-scents ]
Bright Citrus Deodorant by Earth Mama | Natural and Safe
for Sensitive Skin, Pregnancy and Breastfeeding,
Contains Organic Calendula 3-Ounce
Price: $10.99
Rating: N.A.
[ Description ]
[ Features ]
[ Reviews ]
[ Buy Now ]

**rephrased observation**:
Progress: Based on **observation** and **previous actions**,
I clicked size option and scent option.
Reasoning: As there is no more options to select and I met
all requirements specified in the goal, next step is to buy the item.
I can focus on:
[ Buy Now ]

Here is the task.

**goal**:
{goal}
**previous actions**:
{previous_actions}
**current observation**:
{obs}

**rephrased observation**:
```

**Prompt for LLM agent**

```
Webshop
Instruction:
i would like a 3 ounce bottle of bright citrus deodorant for sensitive
skin, and price lower than 50.00 dollars
[ Search ]

Action: search[3 ounce bright citrus deodorant sensitive skin]

Observation:

Progress: I searched the keyword '3 ounce bright citrus deodorant
sensitive skin' to see the relvant items, And now I am looking at the
item list.
Reasoning: Based on the Progress and current observation, the
next step is to choose an item satisfying the specification.
I can focus on:
[B078GWRC1J]
Bright Citrus Deodorant by Earth Mama | Natural and Safe for Sensitive
Skin, Pregnancy and Breastfeeding, Contains Organic Calendula 3-Ounce
$10.99
[B078GTKVXY]
Ginger Fresh Deodorant by Earth Mama | Natural and Safe for Sensitive
Skin, Pregnancy and Breastfeeding, Contains Organic Calendula 3-Ounce
$10.99

Action: click[B078GWRC1J]

Observation:

Progress: I searched and and clicked the item seems to be most
relevant to the goal specification.
I am looking at the option list.
Reasoning: As the goal requires 3-ounce bottle, I can focus on
the size option.
I can focus on:
size
[ travel set (4-pack) ]
[ 3 ounce (pack of 1) ]
[ 3-ounce (2-pack) ]

Action: click[3-ounce (pack of 1)]

Observation:

Progress: I searched and and clicked the item id.
Among the option list, and I clicked size option.
Reasoning: According to the progress, I have to focus on the
scent option as a next step.
I can focus on:
scent
[ assorted scents ]
[ bright citrus ]
[ calming lavender ]
[ ginger fresh ]
[ simply non-scents ]

Action: click[bright citrus]

Observation:

Progress: Based on **observation** and **previous actions**, I clicked
size option and scent option.
```

```
Reasoning: As there is no more options to select and I met all
requirements specified in the goal, next step is to buy the item.
I can focus on:
[ Buy Now ]

Action: click[Buy Now]

Now Here is the task.

Instruction:
{instruction}

{History of observations and actions}

Observation:
{observation}

Action:
```

## C.3 ACTION MATCHING EVALUATION PROMPT

We present the prompt formats for the action matching described in Algorithm 1.

**Prompt for model-based evaluation of action matching**

```
<system prompt>
Your task is to evaluate whether the given two action commands are
semantically aligned.

<main prompt>
You will be given
1). **reference action** which indicates an correct action.
2). **predicted action** which is predicted by assistant agent

Your task is to assess whether the message in **predicted action**
is semantically aligned with message in the **reference action**.
Please make sure you read and understand these instructions
carefully.
Please keep this document open while reviewing, and refer to it as
needed.

Evaluation Criteria:
Alignment = 1: the predicted action is semantically aligned with the
reference action.
    send_msg_to_user('30%') and send_msg_to_user('The percentage of
    amount of pending orders among entire orders is 30%') are
    semantically aligned.
    click('a34') and click('a34', button='left') is semantically
    aligned.

Alignment = 0: the predicted action is semantically not aligned with
the reference action.
    send_msg_to_user('$25') and send_msg_to_user('The requested
    value is $29') are not semantically aligned.
    click('a34') and click('a34', button='left') are semantically
    aligned.

Evaluation Steps:
1. Write a simple feedback that assess whether the predicted action
is semantically aligned with the reference action.
2. After writing a feedback, write a score that is 0 or 1. You
should refer to the Evaluation Criteria.
3. The output format should look as follows: "Feedback: (write a
```

```
feedback for criteria) [RESULT] (an integer number among 0 or 1)"
4. Please do not generate any other opening, closing, and
explanations.

**reference action**: {ref_action}
**predicted action**: {pred_action}

Feedback:
```

