# OpenReview forum: "Learning to Contextualize Web Pages for Enhanced Decision Making by LLM Agents"
_ICLR.cc/2025/Conference — ICLR 2025 Poster_

### Official Review · Reviewer_628v · 2024-10-22

**Soundness:** 3
**Presentation:** 3
**Contribution:** 3
**Rating:** 8
**Confidence:** 2

**Summary:**

The paper proposes training a language model to contextualize complex web pages for improving the success rates of LLM-based web agents. To enable this the proposed method uses the web simulator environments to roll out multiple trajectories and uses multiple LLMs to score the different candidates. This strategy provides a significant improvement over baseline open and closed-sourced models across different benchmarks like WebShop and WebArena.

**Strengths:**

Pipelines for LLM-based web agents are complex and the proposed approach breaks down "contextualizing" the web pages separately from decision making ability of the agents. The approach of leveraging simulation data across the web environments to train (fine tune) a small model for contextualizing shows good results. The reward model is in essence a LLM-based judge system across multiple powerful LLMs. The qualitative results are interesting as they highlight how the proposed contextualization module works in removing irrelevant components of the web page.

**Weaknesses:**

* With respect to generalization capabilities,  the study  can be strengthened by demonstrating performance across the different web environment bechmarks or types of web pages (e.g, instead of picking or holding out 500 examples randomly the type of Web tasks could be  used for creating the train/test held out set).

* Additionally, it is not clear if in the real world environment a simulation environment is available to bootstrap and roll out the candidate sequence of state and action(s).  As listed in the limitations section the power/promise of these agents diminishes given that performance drops when dealing with unseen UI elements.

* For different web benchmarks, different LLMs were used for training the contextualization module. This needs to be explained or justified.

**Questions:**

* Explain the performance of different choices of LLM-based contextualization modules
* Discuss the practical efficacy of these said agents given the cost/tokens for using the reward models and the lack of simulation environment for new unseen web sites.

---

> ### Author Response · Authors · 2024-11-21
>
> Dear Reviewer 628v,
> We sincerely appreciate your efforts and thoughtful comments to help improve our manuscript. Below, we provide detailed responses to each of your comments.
>
> ---
>
> **[W1] Generalization across web environments**
>
> Thank you for the constructive suggestion. Following your comment, we additionally conducted two experiments to evaluate the generalization of LCoW.
>
> First, we evaluated generalization to different task types in the WorkArena benchmark.
>
> Among the 33 task types in the WorkArena, we selected 13 task types as seen task types, and the remaining 20 task types as unseen. We trained the contextualization module using samples collected from the 13 task types and evaluated the contextualization module on the 100 tasks corresponding to the remaining 20 task types.
>
>  As shown in the table below, the contextualization module trained via LCoW generalizes even for task types that were unseen during training, demonstrating generalization between different task types is feasible.
>
> |  | GPT-4o | Gemini-1.5-flash |
> |-----------------------------------------------------|------------|------------------------|
> | Raw observation                                   | 25%       | 10%                    |
> | LCoW                                                    | 30%       | 26.8%                |
>
> Secondly, we have considered a new benchmark during the rebuttal, namely WebArena-lite [1]. We chose this benchmark because it consists of hundreds of task types and 6 websites yet is also compact (165 evaluation tasks), thus enabling us to run it during the short rebuttal period. As shown in the table below, LCoW outperforms the baseline, demonstrating that LCoW can also be applied to general web browsing tasks.
>
> | Webarena-lite  | GPT-4o   |
> |-------------------------------------------|--------------|
> | Raw observation                        | 29.7%    |
> | LCoW                                         | 35.8%    |
>
>
> ---
>
> **[W2] Availability of extension to real web environments**
>
> We would like to highlight that LCoW has the potential to be extended to real-world web environments where predefined goals and corresponding task rewards are not available. Recent research [1] introduced the outcome reward model (ORM), which allows for obtaining rewards based on arbitrary goals and web browsing trajectories. By synthesizing diverse goals and rolling out the agent in an open-ended web environment, successful trajectories can be collected using the ORM. Expanding the training environment of LCoW to real-world websites is an exciting direction for future research.
>
> [1] WebRL: Training LLM Web Agents via Self-Evolving Online Curriculum Reinforcement Learning (2024)
>
> ---
>
> **[W3, Q1] Justification on different LLM backbones for each benchmark**
>
> Firstly, due to differences in task complexity between WebShop and WorkArena, we used a 3.8B-scale model for WebShop and an 8B-scale model for WorkArena. Specifically, WebShop operates within a simulated web environment with a simplified observation space compared to real-world web environments, where token lengths corresponding to single web page observation is lower than 1,000. In contrast, WorkArena involves a real-world web environment with a more complex observation space, where web page observations longer than 10K are prevalent.
>
> We also observed that a relatively smaller model (i.e., Phi-3-mini) tends to memorize the contextualization data rather than effectively learning to contextualize lengthy web pages. Specifically, this was evidenced by a common sign of overfitting: the loss value exhibited a stepwise decrease throughout the training epochs, indicating that the model was fitting the training data too closely without generalizing effectively.
>
> Given these issues, we determined that a larger model with a greater capacity is necessary for the contextualization module to handle the complexity of real-world web browsing benchmarks such as WorkArena and WebArena-lite.

---

> ### Author Response · Authors · 2024-11-21
>
> **[Q2] Practical efficacy of LCoW regarding cost and lack of simulation environment**\
> Firstly, we would like to emphasize that LCoW does not require collecting large-scale contextualization data. Specifically, we collected less than 2,500 samples to train the contextualization module in both WebShop and WorkArena experiments. Moreover, even if the scale of contextualization data collection were increased to train general-purpose web contextualization modules, we believe LCoW would remain feasible. This is supported by the decreasing costs of closed-source LLMs and the emergence of free LLM APIs, which make such efforts increasingly cost-effective.
>
> Secondly, the web itself can serve as a simulation environment for arbitrary websites. While it does not provide specific goals or corresponding rewards, recent studies [1] have utilized LLMs as reward functions, and others have trained outcome reward models specifically for web browsing [2]. By leveraging these methods, agents can collect trajectories from open-ended websites, label their success or failure, and effectively train in real-world web environments.
>
> [1] Pan et al., Autonomous Evaluation and Refinement of Digital Agents (2024)\
> [2] Qi et al., WebRL: Training LLM Web Agents via Self-Evolving Online Curriculum Reinforcement Learning (2024)

---

> ### Author Response · Authors · 2024-11-25
>
> Dear Reviewer 628v,
>
> Thank you again for your time and efforts in reviewing our paper.
>
> As the discussion period draws close, we kindly remind you that two days remain for further comments or questions. We would appreciate the opportunity to address any additional concerns you may have before the discussion phase ends.
>
> Thank you very much!
>
> Many thanks,
> Authors

---

> > ### Comment · Reviewer_628v · 2024-11-26
> >
> > Thank you for the rebuttal. My ratings are aligned with the review and the paper.

---

> > > ### Author Response · Authors · 2024-11-27
> > >
> > > We are happy to hear that our response resolved your concerns. If you have more questions, please feel free to discuss them with us. Thank you once again for your invaluable feedback and the thoughtful effort you invested in reviewing our paper.

---

### Official Review · Reviewer_iKgX · 2024-10-28

**Soundness:** 2
**Presentation:** 2
**Contribution:** 2
**Rating:** 5
**Confidence:** 4

**Summary:**

The paper proposes to contextualize the observation of LLM-based online shopping agents to improve their performance. It trains a task-observation-reliant contextualization module to help locate the most important information on a page and provides explanations. The idea is clever and shows promising results on two shopping benchmarks. However, it doesn't include code or any playing episodes for the reviewers to verify the outcomes.

**Strengths:**

- The observation contextualization idea is clever, and the training of the contextualization doesn't require human-labeled data.
- The results reported on the shopping tasks are promising, proving that the idea should work.

**Weaknesses:**

- I like the observation contextualization idea, but I've seen a highly ranked paper on the WebArena benchmark, a benchmark with a wider type of websites defined other than mere shopping here, using a similar but more general method that doesn't require task-related inputs. I believe the strong reliance on the web observation's format, as you mention in the limitation section, "it often struggles to provide suitable contextualization for web pages containing UI elements unseen during the training," constrains this work's scope on shopping-related tasks only.
- I would question your results as you didn't include code or episodes for reviewers to verify your conclusions, especially when you only play your agents on a partial WorkArena benchmark, whose results are easily controllable if you only select the tasks where your agents win.
- I think you have sacrificed the agent's generalizability with the contextualization module specifically trained on the shopping tasks, as you put in the appendix.

**Questions:**

- How do you know the performance gain is enhanced by the LLM's decision-making?
- Figure 1: How do you select the reported tasks from WorkArena?
- How do you ensure that crucial elements are not removed during the contextualization?
- It seems the contextualization module could only be trained with successful trajectories from LLMs. What about those tasks that even those LLMs could not fulfill?
- The WorkArena contains up to 1,000 task instances, why do you evaluate only 115 tasks? How do you select them?
- If the contextualization is shopping relevant, do you believe it's less convenient to write several human oracle rules than to train a contextualization module?
- Figure 8: The mean of the number of steps doesn't seem to differ much. What is the exact number?
- It seems the behavior cloning baseline is trained with fewer demonstrations than the contextualization module.

---

> ### Author Response · Authors · 2024-11-21
>
> Dear Reviewer iKgX,
> We sincerely appreciate your efforts and thoughtful comments to help improve our manuscript. Below, we provide detailed responses to each of your comments.
>
> ---
>
>
> **[W1, W3] Limited scope of training and evaluation benchmarks**
>
> Thank you for your comment on the scope and generalizability of our contextualization module. To address your concern, we clarify that our experimental evaluations extend beyond shopping-related tasks. The WorkArena benchmark, as highlighted by reviewer xrwv, includes diverse tasks such as information retrieval from dashboards, form completion, and knowledge base searches, among others.
>
> To further assess the generalization capability of LCoW, we evaluated its performance on the WebArena-lite benchmark [1], a compact yet diverse collection of websites (e.g., Gitlab, Map). We trained the contextualization module on the training split of WebArena and measured the success rate on the 165 tasks in WebArena-lite. As shown in the table, LCoW results in a noticeable improvement in performance, highlighting its effectiveness in handling diverse tasks across various websites. Experimental details and additional results are provided in Appendix A.1 of our revised manuscript.
>
> | Webarena-lite evaluation tasks | GPT-4o   |
> |-------------------------------------------|--------------|
> | Raw observation                       | 29.7%    |
> | LCoW                                        | 35.8%    |
>
> [1] Xiao L., et al., “VisualAgentBench: Towards Large Multimodal Models as Visual Foundation Agents” (2024).
>
>
> ---
>
> **[W2]  Code & output trajectories for verification**
>
> In the revised supplementary material, we have included anonymized code and trajectory files corresponding to the results presented in Tables 1 and 2 of our manuscript. Furthermore, we will open-source the uploaded code and model checkpoints to ensure that our findings can be independently verified.
>
> ---
>
> **[Q1] Performance gain enhanced by LLM’s decision making?**
>
> Our main hypothesis is that the primary bottleneck for LLM-based web agents lies in understanding complex web page observations, rather than in the decision-making capabilities of the LLMs themselves. This is because decision making in web browsing is relatively simple compared to tasks such as advanced coding or mathematics, where LLMs typically excel. To validate this hypothesis, we compared the performance of the Gemini-1.5-Flash agent operating on raw observations with that of the same agent utilizing web page observations contextualized by GPT-4, as illustrated in Figure 1. The results clearly demonstrate that contextualizing web page observations significantly enhances performance. These findings motivated us to develop a module specifically designed to contextualize complex web page observations.
>
> ---
>
> **[Q2, Q5] Why and how did authors select a subset from WorkArena for the evaluation?**
>
> Firstly, as for Figure 1, we selected the first 40 tasks from the initially chosen set of 115 tasks for the proof-of-concept experiment. We have clarified this in the revised PDF version.
>
> Secondly, we would like to clarify that using a subset of tasks is a common evaluation setup for WorkArena [1]. For instance, the original benchmark paper also evaluates performance on a subset of the full task set. This is mainly due to our limited budget to cover the inference cost. To ensure fair evaluation, we followed the original paper’s recommendation by selecting an equal number of individual tasks from each task type.
>
> However, for a more comprehensive evaluation, we have added 50 additional tasks to our evaluation and report the results on a total of 165 tasks on 4 types of LLM agents. As shown in the table below, LCoW continues to achieve a higher success rate than baseline agents, underscoring the robustness of its performance. We have updated Table 2 in the manuscript.
>
>
> | 33 task types x 5 seeds | Claude  | GPT-4o  | Gemini-flash | Llama 3.1-70B |
> |---------------------------------|------------|-------------|-------------------|---------------------|
> | Raw observation            | 44.8%   | 38.2%     | 11.5%            | 26.1%            |
> | LCoW                             | 55.7%   | 44.2%     | 41.2%           | 40.0%            |
>
> [1] WorkArena: How Capable Are Web Agents at Solving Common Knowledge Work Tasks? (2024)

---

> > ### Comment · Reviewer_iKgX · 2024-11-23
> >
> > **[W1, W3] Limited scope of training and evaluation benchmarks**
> >
> > The concern is, how well does your contextualization model generalize to other websites **unseen during training**? As you've mentioned, you need to train the model for the WebArena benchmark, then how well does it play on WebShop? What about real-world websites (like the benchmark proposed in [WebVoyager](https://arxiv.org/abs/2401.13919))? Training the module requires LLM calling for collecting training trajectories. I doubt that you find a poor balance between effectiveness (your work has not exceeded SOTA on WebArena) and cost (sunk cost for collecting training trajectories).

---

> > > ### Author Response · Authors · 2024-11-24
> > >
> > > **[W1, W3] Limited scope of training and evaluation benchmarks**
> > >
> > > To address the reviewer's concerns about generalization, we conducted additional experiments with unseen websites. The WebArena benchmark includes tasks across six websites (GitLab, CMS, Reddit, Map, Wikipedia, and Shopping). We trained our contextualization module using data from five of these websites, excluding Shopping, and then tested on this held-out site. The results, displayed below, show that our method outperforms raw observations on this unseen website, which supports its generalization capability.
> > >
> > > |           Unseen website (Shopping)        | GPT-4o |
> > > |-----------------------------------------------------|------------|
> > > | Raw observation                                   | 17.4%       |
> > > | LCoW                                                    | 21.7%       |
> > >
> > > Regarding your concerns about SOTA performance, our main contribution is demonstrating the significant impact of contextualization on enhancing LLM decision-making abilities (highlighted as a strength by Reviewer 628v) with practical compute expenses (as highlighted by Reviewer mbv1). While our model may not yet achieve production-level, commercial SOTA performances on all tested benchmarks, our research provides clear findings that offer new insights into and demonstrate the potential of 'contextualization' in LLM agents. Additionally, since our method is designed to complement existing decision-making strategies, which typically focus on RL fine-tuning or extensive prompting, it has the potential to be integrated with current SOTA methods to further enhance outcomes.

---

> > ### Comment · Reviewer_iKgX · 2024-11-23
> >
> > **[Q6] Necessity of LCoW on Web shopping tasks**
> >
> > As the functionality of a website is limited and could be decomposed into several atomic actions/functions, I think you deserve to know the previous SOTA on the WebArena benchmark [SteP](https://arxiv.org/abs/2310.03720) on how it designs policy for those websites. It's cost-effective.

---

> > > ### Author Response · Authors · 2024-11-24
> > >
> > > **[Q6] Necessity of LCoW on Web shopping tasks**
> > >
> > > STeP proposed a stacked policy model based on human-written subroutines, where the stacked policy efficiently accomplishes given tasks by flexibly composing policies in charge of each subroutine. Especially, STeP might be effective in the limited task domain like web shopping, which requires only a small number of hand-crafted subroutines. However, it also has a limitation of requiring manually defined subroutines (human-written workflows) for web tasks as highlighted in [1]. In contrast, our LCoW is designed to be applicable across a broader range of web tasks. Unlike STeP, LCoW does not require the creation of numerous hand-crafted subroutines, offering a more flexible and scalable approach.
> > >
> > > [1] Kapoor et al., AI Agents That Matter (2024)

---

> ### Author Response · Authors · 2024-11-21
>
> **[Q3] Crucial elements can be removed during contextualization**
>
> While contextualization might remove or modify important parts of the original web observation, our framework mitigates this through the use of the action-matching rewards. Specifically, the rewards encourage the preservation of crucial elements in the contextualized observation, as omitting such elements would lead to incorrect action predictions by the LLM agents. Since LCoW trains a model to generate contextualized observations that maximize the action-matching reward, the resulting contextualizer is naturally inclined to retain essential information.
>
> ---
>
> **[Q4] Reliance on successful trajectories**
>
> We note that while training during a single iteration of LCoW relies on a fixed set of successful trajectories, the number of collected trajectories increases across iterations in the WebShop benchmark. For example, in the first iteration (LCoW-iter 1), we collected 220 successful trajectories for 500 training tasks. By the second iteration (LCoW-iter 2), this number increased to 280 successful trajectories across the entire training set.
>
> Moreover, the iterative trajectory collection process can be further improved by incorporating search methods [1,2], which can enhance the training of the contextualization module.
>
> [1] Language Agent Tree Search Unifies Reasoning Acting and Planning in Language Models, Neurips (2023)\
> [2] Tree Search for Language Model Agents, 2024\
>
> ---
>
> **[Q6] Necessity of LCoW on Web shopping tasks**
>
> We would like to first clarify that the contextualization capabilities of LCoW are not limited to web shopping tasks but extend to a wide range of tasks, as demonstrated on the WorkArena benchmark. Furthermore, we believe that implementing rule-based systems is particularly challenging, even in more constrained domains like web shopping, due to the highly diverse and nuanced nature of human requests. For example, consider a user request like: "Find a lightweight jacket from a sustainable brand that costs less than $100 and has at least 4 positive reviews." It is difficult for rule-based systems to effectively handle nuanced expressions such as "sustainable brand" or "has at least 4 positive reviews," which require deeper contextual understanding. Thus, LCoW’s approach to contextualization offers a more flexible solution, capable of handling complex and varied task types beyond just web shopping.
>
> ---
>
> **[Q7] Behavior cloning baseline seems to use fewer training demonstrations than LCoW**
>
> We would like to clarify that both LCoW and the behavior cloning baseline were trained using the same number of demonstrations: a total of 264 demonstrations with 1594 observation-action samples. In the revised manuscript, we have reported the number of demonstrations (rather than observation-action pairs) for consistency.
>
> ---
>
> **[Q8] Clarification on Figure 8: Exact average values**
>
> The mean action steps for Llama-3.1-70B (raw observation) and LCoW+Llama-3.1-70B are 6.21 and 4.82, respectively. Additionally, the mean action steps for Claude-3.5-Sonnet (raw observation) and Claude-3.5-Sonnet + LCoW are 5.67 and 5.26, respectively.
>
> Additionally, we would like to emphasize that among the 33 tasks successfully completed by both LCoW+Llama-3.1-70B and Llama-3.1-70B, the cases where LCoW+Llama-3.1-70B succeeded in fewer steps were four times more frequent, supporting our claim that the contextualization aids efficient decision making.

---

> ### Comment · Reviewer_iKgX · 2024-11-23
>
> **[Q2, Q5] Why and how did authors select a subset from WorkArena for the evaluation?**
>
> How did you select 115 tasks for the proof-of-concept experiment? What about the other 50? What is the baseline's performance (better than raw observation but how well does your agent equipped with the contextualization module compared with SOTA)? Is it the best performing one in the same track (domain adapted module/observation adaptation)?

---

> > ### Author Response · Authors · 2024-11-24
> >
> > **[Q2, Q5] Why and how did authors select a subset from WorkArena for the evaluation?**
> >
> > As described in line 322 of the manuscript, we selected five instances for each of 23 task types, totaling 115 tasks. Additionally, in response to the review process, we incorporated an extra 50 tasks by selecting five instances from each of 10 additional task types previously unexplored.
> >
> > Regarding the baseline's performance, no previous models have been trained in WorkArena, so we established a behavior cloning baseline for comparison. Our method outperforms this baseline, as demonstrated in Figure 9 of the manuscript.

---

> ### Comment · Reviewer_iKgX · 2024-11-23
>
> **[Q4] Reliance on successful trajectories**
>
> I think the program chair should have made great efforts to ensure that reviewers have been working on the same/relevant topic as the assigned paper. My question was about "those tasks that even those LLMs could not fulfill." From my experience, the web agent has a probability of completing a set of similar tasks (e.g., tasks instantiated with the same intent). Let's say 10% at the beginning. If you look into the success rate details, the training likely improves that likelihood, for instance, increasing from 10% to 15%. But for those tasks that the LLM could not fulfill, the starting probability is 0%. There's little chance, if any, for the agent to complete those tasks w/o golden answer data, either from human labelers or from more performant models. Your training pipeline excludes humans, which I think is one of the methodology advantages you claim, rendering it almost impossible to increase those tasks' success rates.

---

> > ### Author Response · Authors · 2024-11-24
> >
> > **[Q4] Reliance on successful trajectories**
> >
> > We would like to clarify that our framework does not exclude human involvement. Optionally, human demonstrations can be integrated, especially for tasks where the LLM initially struggles. Specifically, the trajectory buffer $\mathcal{T}$ in Algorithm 1 can be initialized as human demonstrations. This allows the model to start with golden answer data, particularly beneficial for challenging tasks where initial success rates are low. Subsequent iterations of training can then leverage these enhanced demonstrations to significantly improve performance. We updated line 259 in a modified manuscript to clarify this point.

---

> ### Comment · Reviewer_iKgX · 2024-12-02
>
> Hi Authors, I have updated the score. There's still lacking evidence that combined with LCoW, models could reach SOTA on a broader range of web tasks (for example, LCoW+previous SOTA>previous SOTA). The remaining concern is the effectiveness when encountering generalizability.

---

> ### Author Response · Authors · 2024-12-04
>
> We sincerely thank the reviewer for the constructive feedback.
>
> First of all, we agree that the generalization to a wide range of tasks is a crucial point for web agents. However, we would like to remark that our method does not hinder generalization, as demonstrated in appendix A.1 and A.2, where generalization to unseen types of tasks and website in real-world web browsing benchmarks (e.g., WebArena-Lite, WorkArena) were achievable even with as few as 2,000 training samples. We believe the generalization capability can be further improved through the expansion of the training data (e.g., utilizing large-scale demonstrations [1] as an initial seed demonstration).
>
> Secondly, adopting LCoW to state-of-art (SOTA) methods might be an interesting direction to be explored and we will include evaluation of LCoW with WebRL, a RL-tuned LLM agent for web browsing tasks, in our final draft. Although the current manuscript does not demonstrate comparison between SOTA and LCoW+SOTA, we believe that our research demonstrates the potential and effectiveness of contextualization for LLM agents.
>
> Thank you once again for actively engaging in the discussion to improve our research.
>
> Many Thanks,
> Authors
>
> [1] Murty et al., NNetscape navigator: complex demonstrations for web agents without a demonstrator (2024)

---

### Official Review · Reviewer_xrwv · 2024-11-02

**Soundness:** 3
**Presentation:** 2
**Contribution:** 2
**Rating:** 5
**Confidence:** 4

**Summary:**

The paper introduces LCoW, a novel framework that addresses the challenge of enhancing decision-making capabilities of Large Language Models (LLMs) in the context of web automation tasks. The method distinguishes the comprehension of web content from the decision-making process by training a dedicated module that creates contextualized representations of intricate web pages.

**Strengths:**

1.The paper presents comprehensive experiments on popular benchmarks, demonstrating that LCoW significantly improves the performance of LLM agents across various scales. The success rates surpassing human experts are particularly impressive.
2.The paper shows that the contextualization module trained with LCoW can generalize well to different LLMs, including those not involved in the training process, which is a strong indicator of the method's robustness.
3.The proposed iterative algorithm for training the contextualization module is a effective approach that allows for continuous improvement of the module based on real-world interactions and feedback.

**Weaknesses:**

1.As mentioned in the paper, the contextualization module struggles with web pages containing UI elements not seen during training. This limitation could be a barrier to the framework's real-world applicability, especially given the vast diversity of web page designs.
2.The contextualization module was trained on a relatively small dataset of fewer than 2,000 self-generated samples. Can the model's ability to generalize to a broader range of web pages and tasks?
3.The paper does not extensively discuss the potential for overfitting, especially given the iterative training process that relies heavily on self-generated data. There is a risk that the model may perform well on similar tasks but fail to adapt to new, unseen scenarios.
4.The contextualization module shown in Figure 7 is not intuitive enough.

**Questions:**

1.The reward obtained from multiple LLMs is only used to judge whether the current step correctly predicts the real action. Should the final expectation of task be used?
2.How does LCoW handle web pages with novel UI elements or layouts that were not encountered during training?
3.Have any measures been taken to prevent overfitting, particularly given the iterative training process that relies on self-generated data?
4.Can the web page be partitioned and analyzed through the prompt function? And there is no comparison with the previous intelligent code analysis work.

---

> ### Author Response · Authors · 2024-11-21
>
> Dear Reviewer xrwv, We sincerely appreciate your efforts and thoughtful comments to help improve our manuscript. Below, we provide detailed responses to each of your comments.
>
> ---
>
> **[W1, Q2] Limited generalization to unseen UI element**
>
> We would like to point out that the challenge of generalization to entirely unseen UI elements or web environments is not unique to our approach. For example, even the recently updated Claude-3.5-sonnet, designed specifically for computer-related tasks, fails entirely on 6 task types corresponding to the “filter-list” category in WorkArena. This is because these tasks demand knowledge of UI elements that are highly specific to the website.
>
> However, we would like to emphasize that difficulty in generalization to completely unseen UI elements does not imply an inability to generalize to unseen task types or web pages. We have conducted a systematic evaluation of LCow’s generalization capabilities on WorkArena. WorkArena features a two-level task hierarchy: categories at the top level and types within each category. The levels of generalization we evaluated are as follows:
> 1) Unseen-type tasks: Tasks of a different type within the same category (i.e., medium generalization)
> 2) Unseen-category tasks: Tasks of a different type and category (i.e., hard generalization)
> For instance, “form filling” is a task \textit{category}, and within it, “creating and submitting an incident report” and “creating new user information” is task \textit{types}. In our experiments, we trained the contextualization module on 13 different tasks and evaluated its performance on 14 unseen-type tasks and 6 unseen-category tasks. Detailed information about the evaluation setup, including the specific categories used, is provided in Appendix A.2 of the revised manuscript.
>
> As shown in the table below, LCoW demonstrated strong generalization to unseen-type tasks, achieving a 22.6% improvement when using Gemini 1.5-flash as the LLM agent. However, we found LCoW to struggle to generalize to unseen-category tasks, highlighting the need for greater task diversity in training or enhanced contextual reasoning to address completely new task types.
>
>
> |                     | GPT-4o              |                               | Gemini-1.5-flash   |                                |
> |------------------|-----------------------|--------------------------|-------------------------|---------------------------|
> |                     | Unseen-type      | Unseen-category   | Unseen-type        | Unseen-category   |
> | Raw observation | 35.7%        | 0.0%                      | 14.5%                  | 0.0%                       |
> | LCoW           | 42.9%               | 0.0%                      | 37.1%                  | 0.0%                       |
>
>
> ---
>
> **[W2] Limited generalization to broader tasks due to limited scale of training data**
>
> We acknowledge the concerns regarding the relatively small dataset of less than 2,000 self-generated samples used to train our contextualization module. The limited size of the dataset was primarily a result of the high computational costs associated with calculating action-matching rewards using closed-source LLMs like Claude, GPT, and Gemini.
>
> Despite these constraints, the LCoW model has demonstrated a significant ability to generalize across a broader range of tasks. To further validate this, we tested LCoW on the WebArena-lite benchmark. This benchmark features a compact yet diverse array of tasks drawn from diverse websites, such as Gitlab, Map, and Reddit, which are representative of different types of web environments.
>
> In our experiments, detailed in Appendix A.1 of our revised manuscript, LCoW showed a substantial improvement in performance on these tasks despite being trained with 2,263 samples. Specifically, as shown in the table below, LCoW achieved a meaningful increase in the success rate across 165 tasks in WebArena-Lite compared to our baselines. This performance enhancement is evidence of the model's robust generalization capabilities, even when trained on a smaller dataset.
>
> | WebArena-lite              | GPT-4o      |
> |--------------------------------|---------------|
> | Raw observation           | 29.7%       |
> | LCoW                            | **35.8%** |
>
> Additionally, as shown in the Table 2, 3 of Appendix A.1, LCoW demonstrates generalization to unseen types of tasks and websites in WebArena-Lite benchmark.

---

> ### Author Response · Authors · 2024-11-21
>
> **[W3, Q3] Potential of overfitting due to iterative training**
>
> We fully agree that overfitting can be a significant challenge when iteratively training the contextualization module. To mitigate this, we have incorporated several components during the iterative training with self-generated data.
>
> First, our iterative process incorporates the collection of new trajectories in each iteration, thus continuously expanding our dataset. For instance, in the WebShop environment, the number of successful trajectories increased from 220 in LCoW-iter 1 to 280 in LCoW-iter 2, enhancing the model's performance on subsequent evaluation tasks.
>
> Second, to ensure robustness against overfitting during the contextualization sampling phase, we employ a technique akin to \textit{rationalization} used in the Self-taught-Reasoner. Specifically, if the action-matching rewards for all sampled contextualizations are zero, we prompt the model with the ground-truth action as a hint. This approach has been shown to collect diverse and high-quality self-generated data, further safeguarding against overfitting. This method has been validated through rigorous testing, demonstrating that our approach not only reduces the risk of overfitting but also improves generalization across unseen scenarios.
>
> [1] Zelikman et al., “STaR: Self-Taught Reasoner” (2022).
>
> ---
>
> **[W4] Update Figure 7**
>
> Thank you for the suggestion. We have revised Figure 7 to make it more intuitive.
>
> ---
>
> **[Q1] Reasons for not using task-level rewards**
>
> We would like to note that task-level rewards were used to select the trajectories for training. Given the sparsity of such task-level rewards, we designed an action-matching reward as a dense training signal for the contextualization module.
>
> ---
>
> **[Q4] HTML parser as an additional baseline**
>
> Thank you for the suggestion. As an additional baseline, we evaluated the recently released Reader-LM, a pre-trained LLM-based HTML parser, and compared its performance to LCoW. As shown in the table below, Reader-LM underperforms compared to the baseline agent. This is primarily because it often repeats the raw observation until reaching the maximum output token limit, rather than effectively summarizing the web page observation. Typical failure cases of Reader-LM have been included in the appendix A.3 of the revised draft.
>
> | WorkArena (165 tasks)    | Success rate |
> |-----------------------------------|-------------------|
> | GPT-4o                             | 38.2%           |
> | Reader LM + GPT-4o       | 9.7%             |
> | Ours                                 | 44.2%           |
>
> Lastly, we would like to note that our LCoW framework can be integrated with parser-based approaches to potentially achieve better performance and generalization. We appreciate your valuable feedback and plan to investigate this direction further, including the results into our manuscript.

---

> ### Author Response · Authors · 2024-11-25
>
> Dear Reviewer xrwv,
>
> Thank you again for your time and efforts in reviewing our paper.
>
> As the discussion period draws close, we kindly remind you that two days remain for further comments or questions. We would appreciate the opportunity to address any additional concerns you may have before the discussion phase ends.
>
> Thank you very much!
>
> Many thanks,
> Authors

---

> > ### Author Response · Authors · 2024-12-02
> >
> > Dear Reviewer xrwv,
> >
> > We greatly appreciate the time and efforts in reviewing our paper.
> >
> > As the discussion period draws close, we kindly remind you that one day remains for further comments or questions. We would appreciate the opportunity to address any additional concerns you may have before the discussion phase ends.
> >
> > Many thanks,
> > Authors

---

### Official Review · Reviewer_mbV1 · 2024-11-04

**Soundness:** 4
**Presentation:** 4
**Contribution:** 4
**Rating:** 8
**Confidence:** 4

**Summary:**

LCoW aims to advance LLM based web agents by adding a contextualization step to the webpage observation in which an LLM reduces the html/raw observation by pruning irrelevant elements and adds contextual descriptive information. This significantly improves the performance of the downstream web agent and sets state of the art results.

The algorithm works by first collecting successful trajectories as ground truth. Then a contextualizer model (with a prompt instructions) is used to reduce and explain the UI elements. This now contextualized observation is give to a set of LLM agents that produce actions. The contextualized model gets a high reward if the agents give the same action as in the successful trajectory. The best contextualized observation is then collected. Finally the model is trained to produce the "good" contextual observations. This can be repeated for multiple iterations.

The main contributions are:
- The novel approach to LLM-based contextualization and parsing, enabling state of the art performance on web agent datasets.
- The algorithm for training the contextualization model
- The prompt for contextualization
- Many experiments

Their results include experiments on
WebShop and WebArena across multiple LLM agents with strong baselines in WebShop (such as AgentQ and LASER).
They also include ablations/analysis on how the action matching reward improves with iterations, how LCoW affects the number of steps required for each task, and comparisons of the original collected trajectories against behavior cloning,

**Strengths:**

LCoW shows a very clear benefit and improvement to LLM web agents.

LCoW shows state of the art improvement on WebShop against strong baselines such as AgentQ and LASER. This is comparable to human expert level on WebShop tasks.

The experiments are pretty comprehensive showing improvement on different benchmarks with different agents and show continuous benefit for up to 3 iterations of LCoW training.

This method does not seem like it would add a ton of compute expenses and could be quite practical.

The paper is also well written.

**Weaknesses:**

There are a few weaknesses of the proposed method, though some of this is more limitations than reasons to reject.

1) It is unclear how this method translates across websites, domains, and to some extent tasks. Since this involves training the contextualization model, there is potential to overfit to the data available. It would be nice to have some experiments showing that LCoW trained on a few domains generalizes to many other domains. Perhaps LCoW when generalizing to a task on say LinkedIn after only being trained on a couple benchmarks.

2) This is also true for generalizing over tasks as well. Perhaps LCoW fails when extending to tasks that require more contextual reasoning.

3) The training details section notes that action matching based on parsing is infeasible for open-ended actions (such as filling in a text box) and uses GPT-4o to do matching. However, the bigger limitation is on open-ended tasks or tasks that have diverse ways/orders of completing them. How would LCoW when there are many actions that are reasonable?

4) In real-world situations there may be many rollouts that include individual actions that are actually incorrect. LCoW would treat these as correct and may even train the model to drop the truly relevant areas of the page.

5) The limitations section only notes novel UI elements as a limitation. It seems the limitation section should be expanded to cover some of the above concerns as well.

6) This approach also relies on being able to collect successful trajectories, whereas other methods that employ search may be able to extend agent capabilities to new tasks.

7) There are no experiments comparing to code-based html parsers for LLM agents. Though they are undoubtably not as good or there would already be models with performance comparable to LCoW.

8) How long does the contextual observation generation take? In tasks that rely on parsing large amounts of text (e.g. Write a tweet based on this article), regenerating and contextualizing a whole article could be expensive, time consuming, and not necessary. (This should be addressed)

There are a few nice to have experiments that are not present:
Generalization across task type or difficulty
Generalization across websites

**Questions:**

Can you add more description of how self-contextualization works? This is the identical contextualization prompt just uses the LLM agent model  (e.g. Gemini-1.5-flash) instead of the trained Phi-3-instruct model.

Table 3 in appendix: Data and caption do not match. 33*15 = 495. Are the numbers the number of successful demonstrations collected? Some more information on the demonstration collection would be helpful.

Does figure 8 include both successful and failed tasks? -> Are the distributions over the same tasks?

Line 182: “select the one that provides the most relevant context for the LLM agent to accurately predict the next action at as the target.”
- This could be written more clearly.

Will the model being released?

---

> ### Author Response · Authors · 2024-11-21
>
> Dear Reviewer mbV1,
> We sincerely appreciate your efforts and thoughtful comments to help improve our manuscript. Below, we provide detailed responses to each of your comments.
>
> ---
>
> **[W1, W2] Generalization to novel tasks and websites**
>
> Following your suggestion, we conducted a systematic evaluation of LCow’s generalization capabilities on WorkArena. WorkArena features a two-level task hierarchy: categories at the top level and types within each category. The levels of generalization we evaluated are as follows:
> 1) Unseen-type tasks: Tasks of a different type within the same category (i.e., medium generalization)
> 2) Unseen-category tasks: Tasks of a different type and category (i.e., hard generalization)
> For instance, “form filling” is a task \textit{category}, and within it, “creating and submitting an incident report” and “creating new user information” are task \textit{types}. In our experiments, we trained the contextualization module on 13 different tasks and evaluated its performance on 14 unseen-type and 6 unseen-category tasks. Detailed information about the evaluation setup, including the specific categories used, is provided in Appendix A.2 of the revised manuscript.
>
> As shown in the table below, LCoW demonstrates strong generalization to unseen-type tasks, achieving a 22.6% improvement when using Gemini 1.5-flash as the LLM agent. However, we found LCoW to struggle to generalize to unseen-category tasks, highlighting the need for greater task diversity in training or enhanced contextual reasoning to address completely new task categories.
>
> |                             | GPT-4o               |                               | Gemini-1.5-flash   |                           |
> |------------------------|------------------------|--------------------------|------------------------|------------------------|
> |                            | Unseen-type        | Unseen-category   | Unseen-type       | Unseen-category |
> | Raw observation | 35.7%                 | 0.0%                      | 14.5%                  | 0.0%                    |
> | LCoW                  | **42.9%**           | 0.0%                       | **37.1%**                 | 0.0%                    |
>
> To further assess the generalization capability of LCoW, we evaluated its performance on the WebArena-lite benchmark [1], a compact yet diverse collection of websites (e.g., Gitlab, Map). We trained the contextualization module on the training split of WebArena and measured the success rate on the 165 tasks in WebArena-lite. As shown in the table, LCoW results in a noticeable improvement in performance, highlighting its effectiveness in handling diverse tasks across various websites. Experimental details and additional results (**generalization to unseen types of tasks, unseen websites**) are provided in Appendix A.1 of our revised manuscript.
>
> | WebArena-lite            | GPT-4o     |
> |------------------------------|---------------|
> | Raw observation         | 29.7%      |
> | LCoW                          | **35.8%** |
>
> [1] Xiao L., et al., “VisualAgentBench: Towards Large Multimodal Models as Visual Foundation Agents” (2024).

---

> ### Author Response · Authors · 2024-11-21
>
> **[W3] Limitations of action matching score: several trajectories exist for a single task**
>
> Thank you for your insightful comment. As you mentioned, there are several open-ended tasks where multiple successful paths exist, but the action-matching rewards alone cannot capture such scenarios when limited to a single successful trajectory. We believe this issue can be addressed by enhancing the trajectory collection strategy to include multiple successful trajectories for a single task. Specifically, integrating search algorithms into the trajectory collection process could help gather diverse successful trajectories, making this an interesting direction for future research.
>
> ---
>
> **[W4] Action matching score: task success does not implies all actions are optimal**
>
> We agree that task success does not guarantee that all actions within trajectory are optimal. However, we would like to emphasize that task success ensures that none of the actions in the collected trajectory are irreversible actions that lead to task failure (e.g., clicking the purchase button before selecting required options in a web shopping task). Therefore, the contextualization module can be trained to perform tasks without failure.
>
> ---
>
> **[W5]  It seems the limitation section should be expanded to cover some of the above concerns as well**
>
> Thank you for your constructive suggestion. Based on your feedback, we have expanded the limitations section in the updated PDF to address the concerns you highlighted.

---

> ### Author Response · Authors · 2024-11-21
>
> **[W6] Reliance on successful trajectories**
>
> It is true that our method requires successful trajectories to train the contextualized module. However, it is important to note that the trajectory collection phase of LCoW can also integrate search methods [1,2] to enlarge the number of successful trajectories in novel tasks. Furthermore, we would like to emphasize that iterative trajectory collection in LCoW increasingly enlarges the number of successful trajectories owing to the improved contextualization module. For example, in the first iteration (LCoW-iter 1) on the WebShop benchmark, we collected successful trajectories for only 44% of the training tasks, and the number increased to 56% in the second iteration (LCoW-iter 2).
>
> [1] Zhou et al., “Language Agent Tree Search Unifies Reasoning Acting and Planning in Language Models.” NeurIPS (2023)\
> [2] Koh et al., “Tree Search for Language Model Agents.” (2024)
>
> ---
>
> **[W7] Additional baseline for parser-based approach**
>
> Thank you for your suggestion. To address your point, we would like to emphasize that the accessibility tree (i.e., HTML cleaned by a rule-based parser) was used as our default raw observation in all experiments. While this approach reduces some noise, the resulting accessibility tree remains verbose and contains redundant content.
>
> To further address your concerns, we have included Reader-LM-1.5B, a recently released LLM-based HTML parser, as a baseline. However, as shown in the table below, Reader-LM demonstrates poor performance as it tends to repeat the input context (web page observation) or sometimes generate non-meaningful texts until it reaches maximum output token limits (2048 tokens), instead of summarizing or rephrasing web page content. We have included examples of the typical failure cases of Reader-LM in the appendix A.3 of the updated PDF version.
>
> | WorkArena (165 tasks) | Success rate |
> |---------------------------------|------------------|
> | GPT-4o                          | 38.2%          |
> | Reader LM + GPT-4o    | 9.7%            |
> | Ours                              | **44.2%**          |
>
> Finally, it is worth noting that our LCoW framework can be integrated with these parser-based approaches, potentially enhancing both performance and generalization.
> Based on your valuable feedback, we plan to further explore this approach and will include these findings in our revised manuscript.

---

> ### Author Response · Authors · 2024-11-21
>
> **[W8] Computational cost for contextualization**
>
> As the reviewer mentioned, we found that contextualization can indeed increase the inference time as it is a generation process. For instance, in the task where the agent has to retrieve specific values or information from a dashboard, a situation similar to that the reviewer pointed out occurs (corresponding charts or tables have to be re-generated). Therefore, we analyzed the wall time of LCoW for 5 tasks related to dashboard retrieval in WorkArena.
>
> In this analysis, the contextualization process took a total of 101.48 seconds on 8 x A6000 GPUs. However, the contextualization process provides an inference time advantage, as the module, implemented using open-source LLMs, summarizes lengthy web page observations. As shown in the table below, the total input token length for the base LLM agent (e.g., GPT-4o) was reduced by 20% through contextualization. Given that web pages exceeding 20K or 30K tokens are common on the web, we believe that the inference time benefit using the contextualization module is more significant.
>
> |                                       | Contextualization wall time | Base LLM cost     |
> |----------------------------------------------------------------|--------------------------------------|-------------------------|
> | Raw observation                                                 | 0 sec                                   | 20,306 tokens      |
> | LCoW                                                                  | 101.48 sec                           | 16,279 tokens      |
>
>
> ---
>
> **[Q1] Details on self-contextualization**
>
> Thank you for your feedback. We used a different contextualization prompt for self-contextualization. We have clarified this in the revised version by including the self-contextualization prompt in Appendix A.4.
>
> ---
>
> **[Q2, Q3, Q4] Error in the manuscript**
>
> Thank you very much for catching these errors.
>
> Regarding the question about Table 3 in the appendix, we would like to clarify that while we attempted to collect successful trajectories from 495 training episodes (i.e., 33 × 15), only 264 successful trajectories were obtained. These 264 trajectories were then used as seed demonstrations. We have updated the caption of Table 9 in the updated PDF version.
>
> Regarding Figure 8, those are the distribution of the number of steps corresponding to the tasks that both LCoW and baseline succeed. We have updated the manuscript for clarification.
>
> Finally, regarding the expression in line 182, we also have modified the expression to make it clearer in the updated PDF file.
>
> ---
>
> **[Q5]  Model release**
>
> We have provided the code and output trajectory files to facilitate reproduction and verification. Furthermore, we will opensource our model checkpoint.

---

> ### Author Response · Authors · 2024-11-25
>
> Dear Reviewer mbV1,
>
> Thank you again for your time and efforts in reviewing our paper.
>
> As the discussion period draws close, we kindly remind you that two days remain for further comments or questions. We would appreciate the opportunity to address any additional concerns you may have before the discussion phase ends.
>
> Thank you very much!
>
> Many thanks,
> Authors

---

> ### Comment · Reviewer_mbV1 · 2024-11-26
> **Reviewer Response**
>
> Firstly, thank you for the response, running these additional experiments, and making the changes to the paper. I think they greatly improve its quality.
>
> I reiterate that I think this research is worth publishing and making the limitations better known improves the paper's quality.
>
> As for the work itself, I think there are some highlighted limitations and weaknesses that are becoming clear.
>
> #1: There are limits to the generalization abilities with out of category tasks getting not performing. Perhaps new categories are more likely to include new types of UI elements. Or perhaps the LCoW learns to filter some types of elements that aren't needed for the training categories but are important for others.
>
> #2: This method adds a very high cost due to regenerating the simplified observation (101 s, generating the context). Since that is per action (any time observation changes), it would make the agent itself incredibly slow. In comparison, a method like the HTML Simplifier from Open Web Agent [1] reduces the size of websites by 99% (in a few milliseconds) and is designed for web agents. May be worth comparing to that, and/or using it as a starting point for LCoW. May be 10 times faster to contextualize 2,000 tokens instead of 20,000 as it could also reduce the number of tokens generated.
>
> [1] Iong, I.L., Liu, X., Chen, Y., Lai, H., Yao, S., Shen, P., Yu, H., Dong, Y., & Tang, J. (2024). OpenWebAgent: An Open Toolkit to Enable Web Agents on Large Language Models. Proceedings of the 62nd Annual Meeting of the Association for Computational Linguistics (Volume 3: System Demonstrations).
> https://aclanthology.org/2024.acl-demos.8.pdf
>
> I reiterate my accept rating and strong all around scores. The contribution score may be a little lower due to the very high latency making this a little less usable in its current form.

---

> > ### Author Response · Authors · 2024-11-27
> >
> > We are glad to hear that our paper has improved based on your constructive feedback. Regarding your comment on limitations, we have clarified this point in the revised version of the PDF. In the final version, we plan to include additional analysis of inference speed. Specifically, we plan to incorporate 1) a comparison of performance and speed with an HTML simplifier and 2) an evaluation of the inference speed improvements achieved through efficient decoding strategies such as speculative decoding. Thank you once again for your thoughtful efforts to enhance our work.

---

### Meta-Review · Area_Chair_sHJH · 2024-12-22

**Metareview:**

This paper proposes the idea of contextualizing web pages to enhance LLM agents' decision-making and introduces a new strategy called LCoW. Reviewers provided diverse opinions on this strategy, with two reviewers holding a relatively positive view and two leaning slightly negative. The main weaknesses raised by the reviewers include: 1) Insufficient discussion of the method's generalization and inadequate experimental comparisons with existing SOTA methods; 2) Certain experimental results, particularly on the WorkArena dataset, were validated only on a subset rather than the full dataset; 3) The method may introduce significant additional computational costs, resulting in efficiency drawbacks, and relies on successful trajectories during training. During the rebuttal phase, the authors addressed some of these concerns, but I think some weaknesses cannot be fully resolved at this stage and should instead be acknowledged as limitations in the paper.
Overall, this is a borderline paper. Taking the reviewers' opinions into account, I am slightly inclined to recommend its acceptance as a poster.

**Additional Comments On Reviewer Discussion:**

During the rebuttal phase, the authors addressed some of these concerns, but I think some weaknesses cannot be fully resolved at this stage and should instead be acknowledged as limitations in the paper.

---

### Decision · Program_Chairs · 2025-01-22

Accept (Poster)